# RELATIONAL KNOWLEDGE DISTILLATION USING FINE-TUNED FUNCTION VECTORS

## ABSTRACT

Representing relations between concepts is a core prerequisite for intelligent systems to make sense of the world. Recent work using causal mediation analysis has shown that a small set of attention heads encodes task representation in in-context learning, captured in a compact representation known as the *function vector*. We show that fine-tuning function vectors with only a small set of examples (about 20 word pairs) yields better performance on relation-based word-completion tasks than using the original vectors derived from causal mediation analysis. These improvements hold for both small and large language models. Moreover, the fine-tuned function vectors yield improved decoding performance for relation words and show stronger alignment with human similarity judgments of semantic relations. Next, we introduce the *composite* function vector - a weighted combination of fine-tuned function vectors - to extract relational knowledge and support analogical reasoning. At inference time, inserting this composite vector into LLM activations markedly enhances performance on challenging analogy problems drawn from cognitive science and SAT benchmarks. Our results highlight the potential of activation patching as a controllable mechanism for encoding and manipulating relational knowledge, advancing both the interpretability and reasoning capabilities of large language models.

## 1 INTRODUCTION

The world and its narratives are composed of objects or words, but humans represent them in a highly structured, relational manner. We excel not just at recognizing instances, but also in understanding and expressing the relations between them to enable analogical inference. This critical ability emerges in the early stages of human development. For example, young children demonstrate creativity by reasoning analogically when answering questions like, "if a tree had a knee, where would it be?" (Gentner, 1977). Adults similarly apprehend analogies for more complex concepts, such as "lawyers are like sharks", "revising manuscripts are like evolving species" or "train is related to track as signal is related to wire". These analogy examples are appealing and interesting for two reasons. First, the two domains involved are semantically distant — for example, a profession (e.g., lawyer) vs. an animal (e.g., shark), or a plant (e.g., tree) versus the human body. In the psychology literature, such cases are typically referred to as *far* analogies(Holyoak, 2025). Second, even when the words or phrases representing a specific relation are not provided in the input text, humans seamlessly extract the relation between entities/concepts and draw inferences accordingly. Indeed, four-term analogy problems, such as "blindness : sight :: poverty : money", provide a core measure of human intelligence and are widely used in educational tests (Gray & Thompson, 2004).

As Spearman (1923) noted, analogy hinges on the ability to extract relations between entities. For instance, solving the analogy "blindness : sight :: poverty : ?" first requires inferring the relation that links the initial word pair (the education of relations), and then integrating this relational knowledge with the meaning of the query word in the second pair to generate a final word such that both pairs instantiate the same relation. Hence, four-term analogy problems can assess two key components of analogical reasoning: extracting the relation between individual entities and transferring that relational knowledge to make inferences.

Large Language Models (LLMs) have demonstrated remarkable advances in showing human-like reasoning abilities in a wide range of tasks (Topsakal & Akinci, 2023; Yan et al., 2024). Regarding

analogical reasoning, however, the evidence is mixed. Some studies indicate that LLMs demonstrate emerging analogical reasoning abilities (Webb et al., 2023; 2025), while other studies indicate that their analogical inferences remain brittle and sometimes produce errors unlike those made by humans (Mitchell & Krakauer, 2023; Lewis & Mitchell, 2024). Although previous studies have evaluated LLMs based on their behavioral performance, none of them have directly examined how relational knowledge is represented internally within these models. It is plausible that LLMs acquire relational knowledge during pretraining on large text corpora. If so, this knowledge could be distilled and strategically leveraged to improve performance on verbal analogy tasks.

In this paper, we focus on LLMs' ability for in-context learning, enabling them to learn tasks "on the fly" from just a handful of examples(Brown et al., 2020; Garg et al., 2022; Furuya et al., 2024; Dong et al., 2022). Without updating its internal weights, the model uses the context provided by the prompt to adapt its behavior. In-context learning in LLMs is inspired by the hypothesis that a model, much like humans, can learn from analogy, generalizing from a small number of examples to temporarily adapt its behavior. However, human analogical reasoning also has a key property that in-context learning in LLMs currently lacks. In addition to using analogy to draw inferences about the task at hand based on relational similarities, humans use this process to acquire more permanent knowledge (Gick & Holyoak, 1983; Holyoak et al., 2024). Whereas in-context learning is transient, humans apply analogies to acquire relational knowledge that can be stored in long-term memory, and which then becomes available for subsequent transfer to novel problems.

By augmenting in-context learning with mechanisms supporting distillation of relational knowledge, we aim to transform relational knowledge in LLMs into representations that can be stored, refined, and used to make inferences. We build on recent work using causal mediation analysis to obtain a compact representation of a specific task, termed the *function vector* (FV). To refine these representations, we propose a fine-tuning algorithm that adapts relation-specific function vectors for relational tasks. We show that *fine-tuned function vectors* (FFV) not only increase performance in relation-based completion tasks, but also show better alignment to human relation similarity judgments. We then show that relation-specific FFVs can be combined linearly to solve analogy problems involving out-of-distribution relations.

## 2 BACKGROUND

**Verbal Analogy in Humans and LLMs.** Although the four-term analogy format, such as "blindness : sight :: poverty : ?", appears simple, task difficulty varies substantially from one problem type to the other. Previous research in psychology shows that humans' success in solving analogy depends on various factors, including the domain similarity between the two word pairs (e.g., "blindness : sight :: deafness : ?" is easier than "blindness : sight :: poverty : ?"), abstractness of semantic relations (e.g., factual relations such as "Beijing : China" vs. abstract relations "loss : grief"), and the degree to which a pair provides a representative instantiation of the underlying relation (e.g., "hot : cold" is a more typical exemplar of an *opposite* relation than "hot : cool") (Holyoak, 2012).

We evaluated four open-source LLMs on two challenging datasets: 40 far-analogy problems from Green et al. (2010) and 374 SAT analogy questions (Turney et al., 2003). See the full list of Green's 40 far-analogy problems in supplemental Table 6. Using a generative prompt in the format "blindness : sight :: poverty : ?", we computed top-5 accuracy, defined as the proportion of problems for which the correct final word appeared among the model's top five generated responses. While the larger models are more capable of solving the four-term analogy problems due to their pretrained knowledge of similar associations, their accuracies on these challenging analogy problems are still 30-40% lower than the easier analogy problems, such as "blindness : sight :: deafness : ?". Specifically we found that the LLMs struggle with analogies from both datasets, with accuracies varying based on model size (i.e. GPT-2 at 22.5% for Green's far-analogy problems, 19.8% for SAT dataset vs. LLaMa-3.1 at 57.5%, 52.7%). See detailed results in Table 9.

**Activation Steering and Function Vectors.** Activation steering, where a vector is used to intervene on an intermediate activation of the model during inference, is a technique often used to manipulate the model's outputs (Turner et al., 2023). A steering vector, extracted from examples of the desired behavior, is added to the hidden states of the model during its forward pass. The model is then expected to generate tokens that align with the steering vector's task. Such vectors can also be

extracted from LLMs directly (Subramani et al., 2022) or updated through either mean activations (Jorgensen et al., 2023) or finetuning (Yin et al., 2024) before being added to the model's hidden states.

Todd et al. (2023) found that a small number (∼10) of attention heads in LLMs serve to encode a compact representation of the relation task demonstrated in the context. An intervention method is developed to identify what they termed function vectors: vector representations of input-output mappings that can be extracted from the LLMs' hidden states based on in-context learning.

First, a causal mediation analysis is conducted to identify a small number of attention heads that contribute to the task representation significantly. Specifically, the LLM is fed a prompt $\tilde{p}_i^t = [(x_i, \tilde{y}_i)]$ where the context sequence is shuffled to create uninformative context, thereby removing the relation consistently shared among word pairs $t$. For each attention head $a_{lj}$ in the LLM, its activation under $\tilde{p}_i^t$ is then replaced by the mean task activation $\bar{a}_{lj}^t$ calculated from the intact context that includes examples consistently reflecting the same task. The increase in word prediction probability is used to calculate its Causal Indirect Effect (CIE), which measures its importance in processing task-relevant prompts:

$$CIE(a_{lj}|\tilde{p}_i^t) = LLM(\tilde{p}_i^t|a_{lj} := \tilde{a}_{lj}^t)[y_{iq}] - LLM(\tilde{p}_i^t)[y_{iq}]$$

Next, the relation-specific function vector is derived by summing up the activity of the top attention heads (typically 10) selected based on the magnitude of their average CIEs.

$$\mathbf{v_t} = \sum_{a_{lj} \in A} \bar{a}_{lj}^t$$

The resulting relation-specific function vector thus serves as the representation of a relation (such as *antonym*), distilled from the implicit knowledge acquired in the pretrained LLM. To test its efficacy, we can now use the relation-specific function vector in a zero-shot test (as shown in the right visual of Figure 1) by injecting it into a LLM during inference time to predict the word that reflects the target relation with the query word (e.g., P("poor"—"rich", $v_{\text{Antonym}}$)).

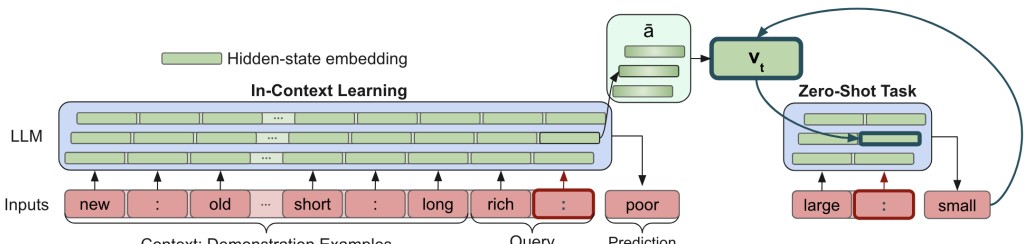

Figure 1: Illustration of in-context learning and zero-shot learning with function vector intervention in LLMs. The context includes a sequence of word pairs instantiating the antonym relation (e.g., *new : old*, *short : long*, etc.). Then, a query word ("rich") is provided and the model needs to predict the next word. If the model predicts "poor" with the highest probability, that will be counted as a correct response under the top-1 criterion of accuracy. The pre-trained LLM's weights remain unchanged during in-context learning, yet some form of learning evidently takes place, as performance on the task is enhanced when demonstration examples are provided.

**Linear Representation Hypothesis.** The linear representation hypothesis states that high-level concepts are represented as linear directions in the model's representational space (Nelson et al., 2021). This hypothesis for language models has been established through the usage of linear probes, causal mediation analysis, and concept edits through targeted projections (Mikolov et al., 2013b; Elhage et al., 2022; Meng et al., 2022). Such representations can be formulated under the mathematical framework of a causal inner product, where separable concepts in a unified space can be represented as orthogonal vectors (Park et al., 2023).

It has been suggested that LLMs can be steered into a particular concept through its representation in the form of a vector. Once the proper vector has been found or constructed, it can be used to guide the

model to process the information based on its target concept (Zou et al., 2023). This is usually done by adding the vector into the LLM's residual stream, which is the basis for activation steering. In addition, if some basis concept representations can be combined linearly to form other concepts, this representation space can be generalized to account for inferences using out-of-distribution concepts.

## 3 METHODS

While the relation-specific function vectors derived through the causal mediation method capture relational knowledge included in the context, we aim to further improve its performance by fine-turning to optimize function vectors using small-sample training. Our framework has two parts: first, we directly update the FV to obtain the fine-tuned function vector (FFV) for a specific relation and examine whether the FFVs capture relation representations aligned better with humans; second, we use linear combinations of FFVs to build the composite function vector (CFV) that represents other relational knowledge for solving analogy problems that are based on out-of-distribution relations.

### 3.1 FINE-TUNED FUNCTION VECTOR

We develop a fine-tuning approach to refine the function vector as a representation of a relation. To start, we use an initial function vector derived from the causal mediation analysis Todd et al. (2023) by contrasting in-context and uninformative context for a specific relation. This function vector is then fine-tuned using a new subset of word pairs that instantiate the relation. The amount of training word pairs per relation can be small, with some relations having as few as 20 word pairs. Given the first word (i.e., query word) in each training word pair instantiating a specific relation and the initial FV for that relation, the training objective is to minimize the difference between the predicted distribution of the response word $y_{pred}$ and the true distribution of the next word that instantiates the relation with the query word $y_{true}$. For backpropagation, we use cross-entropy loss $CE(y_{pred}, y_{true})$ to fine-tune the relation-specific FV. The final loss $L_z$ incorporates this alongside a weighted L2 regularization term:

$$L_z = CE(y_{pred}, y_{true}) + \lambda ||\mathbf{v_z}|| \tag{1}$$

which is then used to update $\mathbf{v_z}$ as shown in Figure 2.

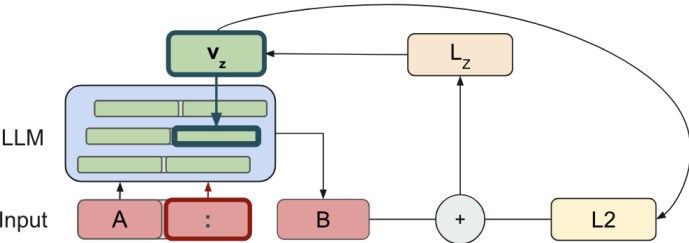

Figure 2: Illustration of the training procedure for the fine-tuned function vector. For a relation $z$, the function vector $\mathbf{v_z}$ is updated using its word pairs (e.g., A : B). The final loss incorporates the cross-entropy from this word pair as well as a weighted L2 norm of the vector for regularization.

It's important to note that the LLM's parameters remain frozen during fine-tuning: only the gradients of the function vector are updated during backpropagation. Because fine-tuning only modifies the values of function vector while maintaining the LLM's pre-trained weights, effective learning can be achieved with a very small training set (e.g., as few as 10 word pairs instantiating the same relation in some of our simulations).

### 3.2 COMPOSITE FUNCTION VECTOR

Function vectors show that implicit relational knowledge in an LLM can be distilled using causal mediation analysis on in-context learning, and can be further refined through fine-tuning. We can consider the approach of fine-tuned function vectors as a way to form explicit representations of

relations. According to the linear representation hypothesis, high-level concepts can be represented linearly in a model's internal representation space (e.g., (Mikolov et al., 2013b; Elhage et al., 2022; Park et al., 2023)). In addition, psychological studies show that humans are sensitive to a set of basic semantic relations that function as core knowledge, and these relations - like other concepts - exhibit typicality gradients that support similarity judgments (Chaffin, 2012; Lu et al., 2019). Neuroscience work further indicates that semantic relations are represented through distinct patterns of brain activation (Chiang et al., 2021). Inspired by these ideas, we propose that fine-tuned function vectors for primary relations can serve as a basis to constructing representations of other relations during inference. The four-term verbal analogy is a standard way to test relational inference with one-shot context (e.g., *furnace : coal :: woodstove : wood*). The first pair, *furnace : coal*, provides the source analog. The second pair, *woodstove : ?*, is the target, which requires applying the source relation to infer the missing word, "*wood*", and thereby complete the analogy.

We developed a *composite function vector* (CFV) for solving one-shot analogy problems involving novel relations that differ from the basis relations in the training pool. This setup constitutes an out-of-distribution evaluation. As shown in Figure 3 a *composite function vector* (CFV) is computed from a source analog, as a weighted sum of relation-specific function vectors.

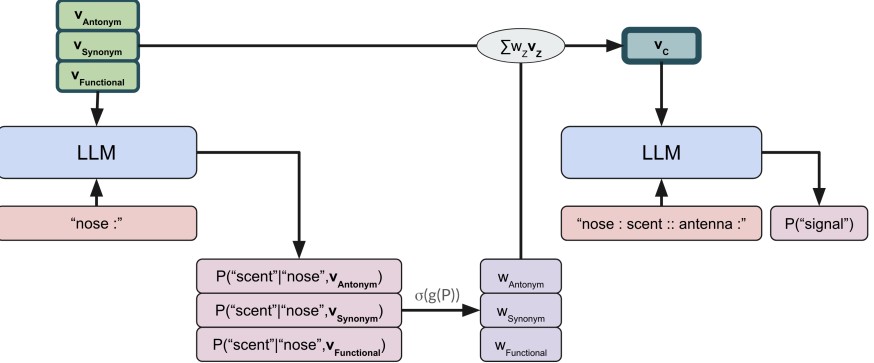

Figure 3: Illustration of the composite function vector for a four-term analogy task (e.g., (*nose : scent :: antenna : ?*). The composite function vector, computed as the weighted sum of the fine-tuned function vectors $\mathbf{v}$ and the posterior distribution $w$ given the source pair (e.g., nose : scent), is injected into the LLM to influence its output predictions for the target. (e.g., (*antenna : ?*).

The first step involves calculating the weights for each of the basis relations, $z$. The weights are determined by the LLM's prediction probability after injecting relation-specific function vectors for a source analog. In other words, LLM's prediction probability indicate the likelihood that a given word pair instantiates each relation. Given a source analog (e.g., (*nose : smell*), we first obtain the probabilities $P(r|q, z)$ of the FFV-injected LLM's output word, $r$, given a query word $q$ and a relation $z$. Afterwards, we apply a softmax $\sigma$ to obtain the probabilities $P(z|q, r)$ of the relation given the source analog.

Within the pool of basis relations, some relations are highly correlated or conceptually similar to others. For example, in our training datasets, the "opposite" relation appears in three datasets (simple-task, SemEval, and MSR), each of which use different sets of word pairs. Rather than manually selecting which basis relations to use, we train an affine transformation $g(x) = Ax + b$ that integrates relation probabilities in a data-driven way to solve one-shot analogy problems. This affine transformation is trained using 2430 analogy problems constructed of word pairs from the same training datasets used to compute the fine-tuned function vectors. To train $g(x)$, we use the same backpropagation method and loss function as for the fine-tuned function vectors $\mathbf{v_z}$.

Combining the steps together, we obtain the weights $w$, summarized by the following equation:

$$w_z = g(P(z|q, r)) = g(\sigma(P(r|q, z))) = A(\sigma(P(r|q, z))) + b \qquad (2)$$

Afterwards, we compute the *composite function vector* (CFV)

$$\mathbf{v_C} = \sum_{z \in Z} w_z \mathbf{v_z} \tag{3}$$

using the resulting weights $w$ from Equation 2, visualized in Figure 3. The resulting CFV amplifies contributions from strong basis relations while attenuating the contributions from weaker basis relations for solving the one-shot analogy problem. This is also used as the basis for training the affine transformation through FV intervention and backpropagation using the loss function in Equation 1.

After determining the relation using the composite function vector for the source analog, we can then transfer this relational knowledge to complete the target analog. Specifically, we inject the CFV into the LLM to solve the analogy for the target (*woodstove : ?*). The relational knowledge from the composite vector can directly interact with LLMs through manipulation of their activations, similar to the functionality of the initial and fine-tuned FVs. Note that the CFV is constructed for a particular word pair; it encodes the relation instantiated between the two words in the source analog and transfers that relational knowledge to guide inference in the target analog.

## 4 EXPERIMENTS

**Model details.** To construct and evaluate our approach, we use GPT-2-Medium, GPT-J-6B, LLaMa-2-7B-Chat, and LLaMa-3.1-8B-Instruct. Each of the models are deployed through Huggingface - details of their architectures are in Appendix Table 2.

### 4.1 DATASETS

**Fine-tuned FV Training.** We used a pool of 118 basis relations, including 15 simple-task relations from the original paper by Todd et al. (2023), 79 relations from SemEval-2012 Task-2 dataset (Jurgens et al., 2012), 8 relations from Google dataset (Mikolov et al., 2013a), and 16 relations from Microsoft Research (MSR) dataset (Mikolov et al., 2013b). The Google and MSR datasets are only used for FFV training to broaden the pool of basis relations by incorporating both semantic and syntactic relations. Additionally, we train 3 complex-task relations from the original paper (Todd et al., 2023) that focus on classification or problem-solving (AG news, CommonsenseQA, Sentiment Analysis) - these are separate from the 118 basis relations in that they are not used for the composite function vector.

The simple-task dataset contains relatively straightforward, factorial relations (e.g., country : capital) and provides varying amounts of word pairs (197-2699) per relation. In contrast, the complex-task dataset consists of relations that involve classifying a sentence to a category or answer (i.e. news headline : news category, question : answer) with a large number of pairs (1167-10962) per relation. The SemEval dataset covers a broad range of abstract relations (e.g., category membership, cause-effect) but offers only 20-30 word pairs per relation. The 79 specific relations are grouped into 10 relation types in SemEval to reflect the hierarchical structure of human relation representations. The Google dataset is similar to the simple-task dataset in that it consists of straightforward semantic and syntactic relations (e.g. adjective : adverb) - however, each relation offers only 20-40 word pairs like SemEval. The MSR dataset only consists of syntactic relations, most of which are counterparts of each other (e.g. adjective : comparative vs. comparative : adjective) and offer 100 word pairs each. The detailed list of relations are included in Tables 4 and 5 at Appendix A.1.

For the simple-task and complex-task datasets, we first apply causal mediation analysis to derive function vectors for each specific relation independently, and then use these relation-specific vectors as initializations for fine-tuning. For each relation, its data was partitioned into three subsets: 1) 70% (138-7673 pairs) for extracting function vectors, 2) 10% (18-987) for fine-tuning them, and 3) 20% (41-2302) for zero-shot testing to evaluate generalization.

For the SemEval dataset, due to the small number of examples per relation, we conduct causal mediation analysis at the relation-type level and use the resulting vectors as initializations for fine-tuning individual relations. Each relation was partitioned into three subsets like for the simple-task dataset, but the finetuning subset always consisted of 10 word pairs while the test subset contained around 3-5 pairs. The remaining pairs were to be grouped with other pairs within its relation type to extract the initial function vector.

**Fine-tuned FV Testing.** For zero-shot tasks that consist of pairs instantiating a specific relation, we use the simple-task and complex-task datasets from Todd et al. (2023), as well as the SemEval-2012 Task-2 dataset (Jurgens et al., 2012). There is no overlap in pairs between the training and testing sets for any of the three datasets.

**Composite FVs.** For one-shot analogy tasks, we use three datasets: Green's analogy dataset (Green et al., 2010), the Bigger Analogy Test Set (BATS) (Gladkova et al., 2016) and the SAT dataset (Turney et al., 2003). Green's dataset includes 80 analogy problems without assigned relation labels; examples include within-domain/semantically near analogies (e.g., *blindness : sight :: deafness : hearing, furnace : coal :: woodstove : wood*) and cross-domain/semantically far analogies (e.g., *blindness : sight :: poverty : wealth, furnace : coal :: stomach : food*). BATS is a large benchmark for analogy task evaluation, consisting of 2000 analogy problems grouped into 40 tasks. Most BATS analogies have clearly defined syntactic or semantic relations within domains, such as regular plurals (*student : students*), participle to past (*following : follows*), member–group (*player : team*), and part–whole (*wheel : car*). Note that there are overlaps in terms of relations used in the training datasets and BATS. The SAT dataset consists of 374 problems from SAT resources, with 90 questions from prep sites, 14 from ETS's site directly, 190 from previous SAT exams, and 80 from guidebooks. With regards to the word pairs and relations, both Green's dataset and the SAT dataset do not have any overlap with the FFV training datasets, while BATS dataset has significant overlaps. Further details on the evaluation tasks is included in Appendix A.2.

### 4.2 FINE-TUNED FUNCTION VECTOR EVALUATION

#### 4.2.1 ACCURACY IN ZERO-SHOT TASKS

For the initial FV, we replicated the results from Todd et al. (2023) in that adding the FV improves the model's performance in the zero-shot tasks. The fine-tuned FVs consistently outperform the initial FVs, yielding increases of around 30-60% for GPT-J (33% for simple-task dataset, 60% for complex-task dataset, 31% for SemEval) and even larger improvements for GPT-2 (45% for simple-task dataset, 61% for complex-task dataset, 35% SemEval). For larger-scale models such as LLaMa-2 and LLaMa-3.1, the FFVs achieve roughly a 20% improvement over the initial FVs. The substantial improvements observed across both small- and large-scale LLMs suggest that FFVs capture relational knowledge more effectively than those derived from causal mediation analysis. Figure 4 shows the average zero-shot accuracies across the simple-task, complex-task, and SemEval relations for GPT-J as an example, while Figure 7 provides the layer-wise accuracies for the simple-task and SemEval relations. Sample results for the LLM baseline, initial FVs, and fine-tuned FVs on the three datasets are provided in Table 7, which reports the average accuracy across 5 seeds.

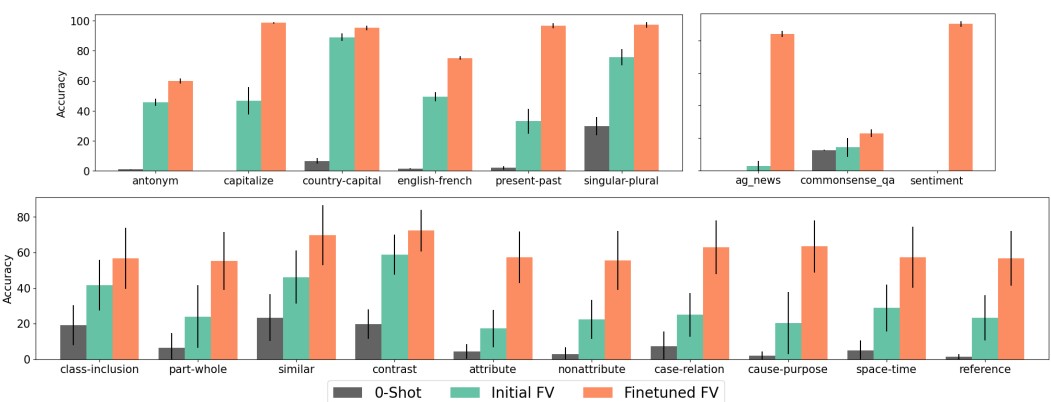

Figure 4: Zero-shot evaluation results for GPT-J: Top-1 prediction accuracy for the 6 representative relations (top left) and problem-solving relations (top right) in the simple-task dataset, and Top-5 prediction accuracy for the SemEval dataset by relation type (bottom). Baseline performance (grey) reflects running GPT-J directly on the Zero-shot task. Compared to the initial FV (green), injecting the fine-tuned FV (orange) into GPT-J yielded the highest average accuracies. The error bars indicate the standard deviation across five runs. FV = function vector.

We also compare the FFVs of the six representative simple-task relations to other activation steering baselines: ActAdd (Turner et al., 2023) and Mean-Centered FVs (Jorgensen et al., 2023). For ActAdd, we use two types of steering vectors - one from using only the basis relation as the prompt $\mathbf{h}_+$, and one from the basis relation along with an opposing term $\mathbf{h}_+ - \mathbf{h}_-$. Using a coefficient of $+5$, we extract the ActAdd steering vectors for the six simple-task relations from Todd et al. (2023) - for example, $\mathbf{h}_+$ for the "singular-plural" relation would be the activations for the prompt "Plural" while $\mathbf{h}_+ - \mathbf{h}_-$ would be the difference in activations for the prompts "Plural" and "Singular". The mean-centered FVs also use the same six simple-task relations from Todd et al. (2023), but on layer 15 as it was deemed to be the best layer for intervention on GPT-J as per the authors. From the results in Table 1, we see that fine-tuned FVs outperform the other baselines even when injected to a later layer at inference time.

| Method | Accuracy |
|---|---|
| GPT-J | 5.1% |
| + $\mathbf{h}_+$ ActAdd (Turner et al., 2023) | 18.0% |
| - $\mathbf{h}_-$ ActAdd (Turner et al., 2023) | 18.5% |
| + $\mathbf{v_t}$ Initial FV | 42.3% |
| + Mean-centering (Jorgensen et al., 2023) | 45.7% |
| **+ $\mathbf{v_z}$ Fine-tuned FV (Trained on Layer 12)** | **75.2%** |

Table 1: Average zero-shot accuracies for GPT-J with interventions on layer 15 compared to other baselines. Compared to GPT-J alone, ActAdd only improved performance by 12.9-13.4%, with only a 0.5% increase from the counterbalanced steering vector $\mathbf{h}_+ - \mathbf{h}_-$ compared to $\mathbf{h}_+$ alone. In terms of FVs, incorporating mean-centering to the initial FVs yielded a 3.4% increase in accuracy - however, this improvement is minimal compared to the 32.9% difference between the initial and fine-tuned FVs. FV = function vector.

Note that, for the finetuning approach, initializing the FVs derived from causal mediation analysis is a key factor for the superior performance of the FFV. The control simulation, where the vector is initialized with random values for the fine-tuning approach, tends to have a weaker performance than the FFV - further details on this are provided in Appendix A.3.

### 4.2.2 RELATIONAL SIMILARITY

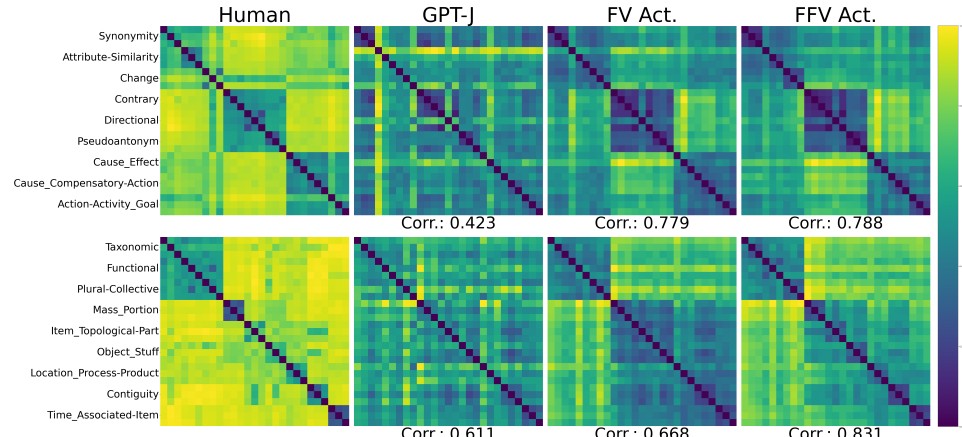

Figure 5: Relational dissimilarity matrices of human judgments and model predictions. Each row and column represents a word pair, and each cell represents the pairwise dissimilarity between the row's word pair and the column's word pair. The warmer the cell's color, the higher its dissimilarity.

The improvement in performance accuracy with FFVs is to be expected with the extra fine-tuning approach. Next, we ask whether the relation representations captured in the FFVs align with human

representations of relations better than initial FVs. Ichien et al. (2022) empirically measured human judgments on relation similarity between word pairs. For example, humans consider the word pair *black:white* to be more relationally similar to *happy:sad* than to *happy:joyful*. To compare the LLM's activity and analyze their relation similarity patterns, we run a dummy relation task where the target query is blank (e.g., *tall : short :: : ?*). We use the same word pairs as the ones used for the two human experiments. For each word pair, we extract the activity of the attention heads in the layer following FV intervention (Layer 13 for our experiments) for the final colon token. By doing so, we can analyze the FV's effects in the model's successive activities.

Figure 5 shows the relational dissimilarity matrices derived from human judgments in the two experiments from the psychological study Ichien et al. (2022), and from an intermediate layer's activations of the baseline (GPT-J), initial function vectors, and fine-tuned function vectors. For this, we extracted the activations in the layer following FV intervention, which was layer 13. Out of the three, the FFV achieved the strongest alignment with human judgments. We computed Pearson correlations between human dissimilarity judgments and the model predictions. Baseline GPT-J showed moderate correlations with human judgments $r = 0.423$ and $0.611$ across the two human experiments, which used different sets of word pairs and relations. The initial FV substantially improved the correlation in Experiment 1 ($r = 0.779$) and produced a modest increase in Experiment 2 (0.668). The FFV achieved the highest correspondence with human judgments in both experiments, yielding $r = 0.788$ in Experiment 1 and a significant improvement to $r = 0.831$ in Experiment 2 relative to the initial FV. The superiority of FFVs in producing human-like relational similarity judgments is evident for both small- and large-scale LLMs. More detailed results are provided in Appendix A.5. In addition, we visualized FFVs of all relations from four datasets. As shown in supplemental Figure 11, there is clear clustering that separates syntactic relations from semantic relations, suggesting that FFVs capture fundamental structures of relational knowledge.

### 4.3 COMPOSITE FUNCTION VECTOR EVALUATION: ONE-SHOT ANALOGY TASKS

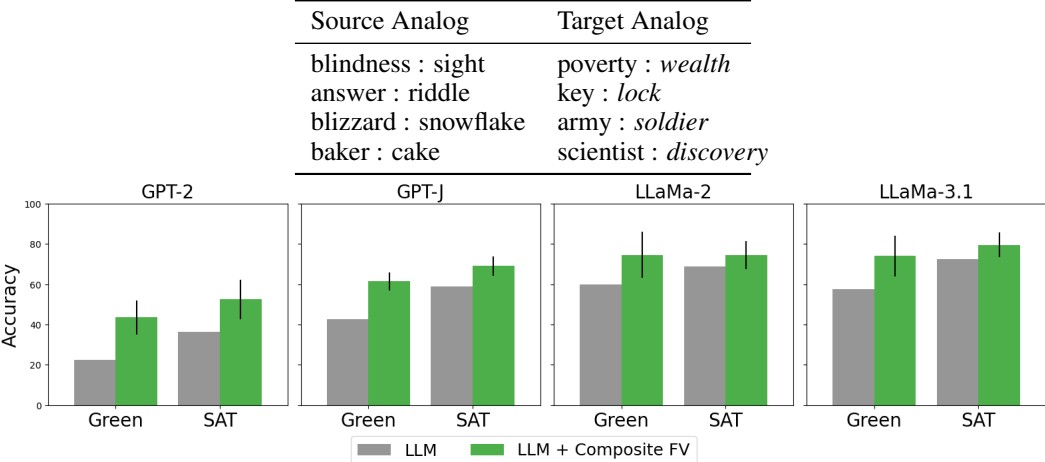

Figure 6: Top: Example analogies from Green's dataset. Italicized words represent the text that is need to be generated by the model. Bottom: Top-5 prediction accuracy for the one-shot analogy problems in Green's far-analogy + SAT datasets. The grey bars represent the accuracies from the pre-trained baseline models, and the colored bars represent its performance after injecting the CFV. The error bars for the CFV represent the standard deviation across five runs. CFV = composite function vector.

We evaluated the composite function vectors using one-shot analogy tasks, such as *blindness : sight :: poverty : ?* Figure 6 provides the accuracy of this task from pre-trained baseline LLMs and with the injection of composite function vectors to these LLMs. For Green's analogy dataset, CFVs did not substantially affect performance on near-analogy problems (left two columns in Table 9). These analogy problems are straightforward (e.g., *furnace : coal :: woodstove : wood*) and considered as an easy task for both humans and LLMs. This is likely because near-analogy problems can be effectively solved by leveraging rich knowledge of semantic associations. When CFVs are used

to steer the LLMs to focus on relational knowledge, alternative strategies based on simple semantic associations may be deemphasized. However, such semantics-based strategies break down for more challenging reasoning scenarios, such as the far analogies involving word pairs from different domains (e.g., *furnace : coal :: stomach : food*).

Critically, composite function vectors yielded significant gains for far-analogy problems in Green's dataset such as *blindness:sight :: poverty: wealth* in which word pairs come from different domains, also called cross-domain analogy. As shown in the left panel of Figure 6, CFVs showed significant accuracy increase by 19% from 49% with the GPT-J baseline to 61% with CFV. This significant improvement from using the CFV for far-analogy problems was consistent across both small- and large-scale LLMs, with increases of 20.5% for GPT-2, 14.5% for LLaMa-2 and 16.5% for LLaMa-3.1. Detailed results are provided in Table 9 of the supplemental materials. These consistent and significant improvements of the CFV across LLMs highlight its robustness in capturing the relational knowledge needed to solve challenging analogy problems, particularly when cross-domain analogies rule out strategies based on semantic associations.

For the SAT dataset, we observe significant improvements from using CFVs on model performance. As shown in Figure 6, the CFV improved performance for all four LLMs on the analogy task (+12.9% for GPT-2, +18.6% for GPT-J, +4.4% for LLaMa-2, +6.1% for LLaMa-3.1). The consistent increases in accuracy across both small-scale and large-scale LLMs highlight the effectiveness of the CFV on analogies consisted of more advanced vocabulary. Even though the FFVs and affine transformation were trained using analogy problems with simple words (i.e. elementary to middle-school vocabulary), the resulting CFV was still capable of completing analogies with less commonly used words (i.e. high-school/university-level vocabulary).

## 5 Conclusions and Discussions

In this paper, we propose that relational knowledge can be distilled from LLMs through the expansion of function vectors. We demonstrate that subsequent finetuning using a small amount of training data not only enhances relational representations (yielding better performance on relation tasks), but also produces human-like similarity judgments for relations. Moreover, fine-tuned vectors can be flexibly combined through linear composition, enabling analogical reasoning over untrained and novel relations. Together, these results showcase the capability of LLMs to develop human-like relational knowledge via more interpretable mechanisms.

Our contributions are threefold. First, we systematically extend and evaluate the function vector approach (Todd et al., 2023) across a wide range of relational tasks, spanning from concrete (e.g., antonyms) and syntactic (e.g. singular-plural) relations to more abstract ones (e.g., causality, category membership). Second, we develop a fine-tuning algorithm that enhances relational representations with minimal training data (e.g., a dozen of word pairs per relation), assessed through both zero-shot task performance and relation similarity judgments against human data. Third, we introduce composite function vectors, which leverage relational knowledge to solve analogy problems involving novel and untrained relations.

While this study provides promising results, it has limitations. One major limitation is that this study's analogies are restricted to four-term problems of the form *A : B :: C : D*. Although this format is the most common in educational testing, analogical reasoning extends far beyond such cases. Future work should expand the framework to analogies between narratives and stories, which involve more complex relational structures. It is feasible to incorporate stories as sentence-level context and apply composite FVs to represent pairwise relations between key entities, thereby supporting analogical reasoning. Another limitation concerns the hierarchical nature of human relational representations. For example, whole–part relations encompass multiple subtypes, such as object–component (*hand : finger*), collection–member (*army : soldiers*), and mass–portion (*hour : seconds*). Further research is needed to determine whether LLMs naturally acquire these hierarchical structures in their relational representations, or whether additional alignment is required to achieve them.

ETHICS STATEMENT

The research in this paper is in accordance with the ICLR Code of Ethics. The methods in this study did not use any personal or sensitive information, as the datasets for both finetuning (Todd et al., 2023; Jurgens et al., 2012; Mikolov et al., 2013a;b) and evaluation (Ichien et al., 2022; Green et al., 2010; Gladkova et al., 2016) are publically available to be used for research. All datasets are used for their intended research purposes.

REPRODUCIBILITY STATEMENT

Details of our experiments, including dataset setup and preprocessing, are provided in the main text and Appendix A.1. The source code is also provided in an anonymous repository.

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

# A   APPENDIX

**Model details**    In this paper, we use GPT-2-Medium, GPT-J-6B, LLaMa-2-7B-Chat, and LLaMa-3.1-8B-Instruct. Each of the models are deployed through Huggingface - details of their architectures are in Table 2.

| Model | Citation | Parameters | $|L|$ | $|a_l|$ |
|---|---|---|---|---|
| GPT-2 | (Radford et al., 2019) | 355M | 24 | 16 |
| GPT-J | (Wang, 2021) | 6B | 28 | 16 |
| LLaMa-2-Chat | (Touvron et al., 2023) | 7B | 32 | 32 |
| LLaMa-3.1-Instruct | (Dubey et al., 2024) | 8B | 32 | 32 |

Table 2: Summary of the models used for this paper. We provide the number of parameters, layers $|L|$, and attention heads per layer $|a_l|$ for each model.

## A.1   EXPERIMENT DETAILS

We tested the LLM and our function vectors (FVs) using a broad range of semantic relations in the SemEval-2012 Task-2 dataset (Jurgens et al., 2012). This dataset, taken from a linguistic taxonomy (Bejar et al., 2012), contains a hierarchical structure of 10 general relation types (e.g., class inclusion, contrast, cause-purpose), each of which consists of 5-8 subtypes for a total of 79 primitive semantic relations. The dataset includes 3,215 word pairs, with 35-48 pairs for each of the 79 subtype relations - examples are provided in Table 3. For each semantic relation, we can compute its corresponding function vector from the LLM's in-context learning. As shown in Table 7, the initial function vector led to a mild improvement in the 0-shot task for SemEval. However, its performance is still lower than 1- and 10-shot in-context learning across all models.

For the composite vector, we used all of the SemEval dataset's (Jurgens et al., 2012) relations as well as tasks from the Google dataset (Mikolov et al., 2013a), MSR dataset (Mikolov et al., 2013b), and the original paper (Todd et al., 2023). We list the remaining tasks in Table 4.

---

**Algorithm 1** Fine-tuned Function Vector Procedure

---

**Input:**

   $\mathbf{v_t} \leftarrow$ Function vector initialized under 10-shot learning for task/relation type $t$
   $D_z = (x_z, y_z) \leftarrow$ Training example for task/relation $z$ of dataset
   A frozen multilayer LLM with forward function $f$
   $\lambda \leftarrow$ A hyperparameter for L2 regularization
1:  $\mathbf{v_z} \leftarrow \mathbf{v_t}$
2:  **for** n epochs
3:      Add $\mathbf{v_z}$ to layer $l$ of LLM
4:      output $\leftarrow$ f($x_z$)
5:      Compute loss $L_z \leftarrow$ CE(output, $y_z$) + $\lambda||\mathbf{v_z}||$
6:      Run backward($L_z$)
7:      Update $\mathbf{v_z}$
**Output:**  A fine-tuned function vector for relation $z$ trained from $D, \mathbf{v_z}$

---

To compute the fine-tuned function vector, we first compute or obtain the initial function vector $\mathbf{v_t} = \sum_{a_{lj} \in A} \bar{a}_{lj}^z$. For $n$ epochs, we then train $\mathbf{v_t}$ into $\mathbf{v_z}$ by adding it to an intermediate layer of the LLM and calculating the loss from there. More details on this process are specified in Algorithm 1.

For training the FFVs, we use an AdamW optimizer with a learning rate of 0.01; the FFVs are trained for 25 epochs in GPT-2, 10 epochs in GPT-J, and 5 epochs in LLaMa-2 and 3.1. For all models, we use layer 12 to inject and train the FFV on as it consistently led to the highest accuracies for the initial FV. By running one word pair per relation in each batch, we can backpropagate on all 79 FVs at once. This was done for 5 seeds, each sampling a new set of word pairs from the relation's pool.

| Type | Relation | Example |
|---|---|---|
| Class Inclusion | Functional
Collective | vehicle → car
clothing → shirt |
| Part - Whole | Object:Component
Mass:Portion
Creature:Possession | face → nose
water → drop
millionaire → money |
| Similar | Synonym
Attribute Similarity
Change | buy → purchase
rake → fork
brighten → color |
| Contrast | Contradictory
Defective | remember → forget
limp → walk |
| Attribute | Object:State
Object:Typical Action | beggar → poverty
soldier → fight |
| Nonattribute | Object:Nonstate
Object:Atypical Action | war → tranquility
recluse → socialize |
| Case Relation | Agent:Instrument
Action:Recipient
Recipient:Instrument | conductor → baton
teach → student
graduate → diploma |
| Cause-Purpose | Cause:Effect
Action:Goal
Prevention | joke → laughter
fertilize → grow
antidote → poison |
| Space-Time | Location:Item
Location:Action
Sequence | bookshelf → books
school → learning
prologue → narrative |
| Reference | Expression
Plan | smile → friendliness
recipe → cake |

Table 3: A subset of tasks and their paradigm examples from each relation type in the SemEval dataset.

Evaluation is done with top-5 accuracy, which measures the proportion of word pairs in the test set for which the model correctly generates the target word as one of the top five predicted responses.

To evaluate model performance, the FFV is injected to layer 12 of the LLM to perform the 0-shot and 1-shot analogy tasks in the test set. This set consists of word pairs that the LLM has never encountered in previous phases of the vector's development (both initialization and optimization) until now. For the 1-shot analogy task, we use the paradigm exemplars in SemEval - these are defined as the word pairs that best resemble the current relation. In conjunction with the pre-trained LLM, the model with the FFV can estimate the probability distribution of the response word. Given a query word $q$ and the FFV $\mathbf{v_z}$ identified for a relation $z$, the prediction distribution $P(r|q, z)$ can be computed by adding the corresponding $\mathbf{v_z}$ to the attention heads' activations $a_l$ in layer $l$ as follows:

$$P(r|q, z) \sim \text{LLM}(r|q, a_l := a_l + \mathbf{v_z}) \tag{4}$$

We also use an AdamW optimizer to train the affine transformation for the CFV's weights; this is trained for 10 epochs with $lr = 0.001$ in GPT-2, 5 epochs with $lr = 0.001$ in GPT-J, and 7 epochs with $lr = 0.0005$ in LLaMa-2 and 3.1. For each task represented in the CFV, we sample 10 word pairs (5 for context, 5 target pairs) for a total of 25 analogies to use for training.

To evaluate the composite function vector, we used four-term analogy problems from the dataset provided by Green et al. (2010), which includes 40 easy (semantically-near) analogy problems (e.g., blindness: sight :: deafness : hearing, furnace : coal :: woodstove : wood), and 40 difficult (semantically-far) analogy problems (e.g., blindness: sight :: poverty: wealth, furnace : coal ::

| Dataset | Relation Name | Example |
|---|---|---|
| Simple-Task | Antonym | flawed → perfect |
| | Capitalize first letter | whose → W |
| | Capitalize | without → Without |
| | Country-capital | Chile → Santiago |
| | Country-currency | Italy → Euro |
| | English-French | those → ceux |
| | Lowercase first letter | WITTY → w |
| | Next-item | zero → one |
| | Previous-item | one → zero |
| | Person-instrument | Skylar Grey → piano |
| | Person-occupation | Billy Roche → actor |
| | Person-sport | Colin Kaepernick → football |
| | Present-past | adapt → adapted |
| | Singular-plural | wallet → wallets |
| | Synonym | curse → swear |
| Google | Adjective-adverb | amazing → amazingly |
| | Comparatives | bad → worse |
| | Male-female | brother → sister |
| | National currency | Argentina → peso |
| | Nationality adjective | Denmark → Danish |
| | Past tense verbs | dancing → danced |
| | Plural verbs | eat → eats |
| | Present participles | code → coding |
| MSR | Adjective-comparative | good → better |
| | Comparative-adjective | better → good |
| | Adjective-superlative | good → best |
| | Superlative-adjective | best → good |
| | Comparative-superlative | better → best |
| | Superlative-comparative | best → better |
| | Noun-plural noun | person → people |
| | Plural noun-noun | people → person |
| | Noun-possessive noun | state → state's |
| | Possessive noun-noun | state's → state |
| | Present verb-Past verb | is → was |
| | Past verb-present verb | was → is |
| | Present verb-Infinitive verb | is → be |
| | Infinitive verb-present verb | be → is |
| | Past verb-infinitive verb | was → be |
| | Infinitive verb-past verb | be → was |

Table 4: The non-SemEval relations used for the composite function vectors and their examples, grouped by dataset.

stomach : food). Accuracy is defined as the proportion of problems in which the correct answer is included among the top-five tokens with highest prediction probabilities.

## A.2 EVALUATION

Our function vectors are evaluated through four tasks:

1. Shuffled-label Evaluation: For each relation, its FFV is evaluated on 10-shot prompts where the context sequences are shuffled. Accuracy is measured with respect to the proportion of word pairs in which the target word is the highest-ranking score. This evaluation task is used to compare the LLM baseline, the FV intervention, and the FFV intervention on the simple-task dataset.

| | Example | |
|---|---|---|
| | **Query** | **Target** |
| AG News | Dogs in Training to Sniff Out Cancer... | Science |
| CommonsenseQA | What home entertainment equipment requires cable?
a: radio shack
b: substation
c: cabinet
d: television
e: desk | d |
| Sentiment Analysis | Very well-written and very well-acted. | Positive |

Table 5: The relations in the complex-task dataset and their examples. These were used to test the efficacy of fine-tuned FVs in relations that involve more complicated formats.

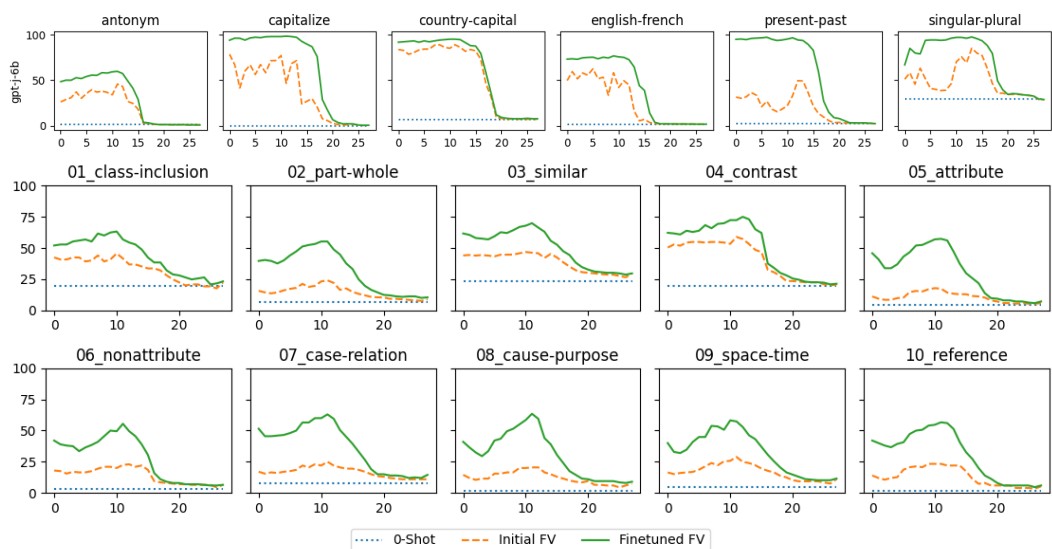

Figure 7: Layer-wise zero-shot results for GPT-J: Top-1 prediction accuracy for the 6 representative relations in simple-task dataset (top row), and Top-5 prediction accuracy for the SemEval dataset by relation type (bottom 2 rows). The fine-tuned FV (solid green) exhibited similar trends to the initial FV (dotted orange) in that injecting the vector in the early-to-middle layers leads to the largest improvements, encouraging the LLM to respond to the prompt with the desired relation. Compared to those of the initial FV, the fine-tuned FV's accuracies across the earlier layers were both higher and more stable, with some relations exhibiting a gradual spike up to layer 12 before dropping. As the FV was fine-tuned based on layer 12 interventions, this may play a small factor in such a trend. FV = function vector.

2. Zero-Shot Evaluation: The models are evaluated on prompts with a zero-shot task given a relation (e.g., "long : ?" + the *antonym* function vector). This evaluation task is used to compare the LLM baseline, the FV intervention, and the FFV intervention on both the simple-task and SemEval datasets. Accuracy for the simple-task dataset is measured in the same way as for the shuffled-label evaluation, while for SemEval it is measured as the proportion of word pairs with target words that are within the top 5 scores.

3. One-shot Analogy Evaluation: Using the source word pair as one-shot context, models predict the target word to complete the analogy (e.g., *furnace : coal :: woodstove : ?*, in which the correct answer is "wood"). Accuracy is measured as the proportion of problems for which the target word is among the top-5 predictions. This evaluation task is used to

compare the LLM baseline and CFV intervention on Green's dataset, BATS, and the SAT problems.

4. Relational Similarity: Using a subset of the word pairs from SemEval, we compare the similarity structure of the relational representations captured by our function vectors to that of human judgments. The human results were provided by Ichien et al. (2022) - the authors utilized the multi-arrangement method to measure human similarity judgments in two experiments (Exp. 1 and Exp. 2 in the original study). Human similarity judgments are used to assess the extent to which word pairs instantiate similar relations. For example, *white : black* and *happy : sad* are judged as similar due to them representing the same type of relation, whereas *white : black* and *happy : joyful* are dissimilar because they reflect different relations.

| Source Analog | Target Analog |
|---|---|
| answer : riddle | key : lock |
| ash : fireplace | lint : pocket |
| aspirin : pain | muffler : noise |
| baker : cake | scientist : discovery |
| basket : picnic | holster : gun |
| basketball : hoop | traveler : destination |
| blindness : sight | poverty : money |
| blizzard : snowflake | army : soldier |
| bracelet : wrist | moat : castle |
| burger : bun | book : cover |
| cleanser : face | absolution : sinner |
| eraser : pencil | amnesia : memory |
| father : son | inventor : invention |
| flock : goose | constellation : star |
| foresight : future | x-ray : bone |
| foundation : house | premise : argument |
| furnace : coal | stomach : food |
| hoof : hoofprint | introduction : impression |
| immunization : disease | forewarning : surprise |
| jacket : zipper | wound : suture |
| ketchup : tomato | fuel : petroleum |
| kitten : cat | spark : fire |
| knee : kneepad | snail : shell |
| lambchop : lamb | chapter : book |
| landscaper : lawn | stylist : hair |
| launchpad : helicopter | divingboard : diver |
| lawschool : lawyer | vineyard : wine |
| movie : screen | lightning : sky |
| multiplication : product | brewing : beer |
| nose : scent | antenna : signal |
| orchard : apple | neighborhood : apartment |
| painting : canvas | birthmark : skin |
| pen : pig | reservoir : water |
| rectangle : perimeter | nation : border |
| revising : manuscript | evolving : species |
| saxophone : jazz | typewriter : poetry |
| sugar : coffee | incentive : deal |
| thermometer : temperature | polygraph : honesty |
| train : track | signal : wire |
| watermelon : rind | cigarette : butt |

Table 6: The word pairs used for Green's far-analogy dataset.

## A.3 FINE-TUNED FUNCTION VECTOR EVALUATION

The sample results for the LLM baseline, initial FVs, and fine-tuned FVs are provided in Table 7 for GPT-J, which reports the average accuracy across 5 seeds. For the simple-task dataset we conducted simulations for the shuffled-label and 0-shot task, while for SemEval we only used the 0-shot task for evaluation due to its limited data per relation.

| | Simple-Task | | Complex-Task | SemEval |
|---|---|---|---|---|
| | Shuffled-Label | Zero-Shot | Zero-Shot | Zero-Shot |
| $[(x_{i1}, \tilde{y}_{i1}), ..., (x_{iN}, \tilde{y}_{iN}), x_{iq}]$ | | $[x_{iq}]$ | $[x_{iq}]$ | $[x_{iq}]$ |
| GPT-2 | $26.46 \pm 3.35\%$ | $1.39 \pm 0.96\%$ | $0.01 \pm 0.02\%$ | $0.65 \pm 1.07\%$ |
| + $\mathbf{v_t}$ Initial FV | $52.64 \pm 2.83\%$ | $16.68 \pm 4.68\%$ | $0.65 \pm 0.78\%$ | $11.05 \pm 8.29\%$ |
| + $\mathbf{v_z}$ **Fine-tuned FV** | $\mathbf{57.43} \pm 3.00\%$ | $\mathbf{61.84} \pm 2.82\%$ | $\mathbf{61.79} \pm 2.40\%$ | $\mathbf{46.42} \pm 16.48\%$ |
| GPT-J | $40.73 \pm 16.86\%$ | $5.13 \pm 1.11\%$ | $35.41 \pm 10.03\%$ | $8.83 \pm 6.63\%$ |
| + $\mathbf{v_t}$ Initial FV | $83.71 \pm 2.33\%$ | $54.11 \pm 5.48\%$ | $44.12 \pm 12.46\%$ | $30.24 \pm 13.56\%$ |
| + $\mathbf{v_z}$ **Fine-tuned FV** | $\mathbf{88.14} \pm 1.56\%$ | $\mathbf{87.22} \pm 1.10\%$ | $\mathbf{63.34} \pm 12.52\%$ | $\mathbf{60.78} \pm 15.49\%$ |
| LLaMa-2 | $29.16 \pm 4.03\%$ | $13.28 \pm 3.27\%$ | $16.86 \pm 0.24\%$ | $8.15 \pm 6.46\%$ |
| + $\mathbf{v_t}$ Initial FV | $82.99 \pm 3.28\%$ | $68.45 \pm 9.11\%$ | $26.03 \pm 16.98\%$ | $42.82 \pm 15.58\%$ |
| + $\mathbf{v_z}$ **Fine-tuned FV** | $\mathbf{86.64} \pm 3.33\%$ | $\mathbf{88.84} \pm 1.85\%$ | $\mathbf{76.02} \pm 4.32\%$ | $\mathbf{65.86} \pm 14.52\%$ |
| LLaMa-3.1 | $18.05 \pm 2.04\%$ | $2.75 \pm 1.59\%$ | $23.67 \pm 0.19\%$ | $11.12 \pm 7.85\%$ |
| + $\mathbf{v_t}$ Initial FV | $71.02 \pm 5.48\%$ | $65.85 \pm 7.49\%$ | $9.8 \pm 6.00\%$ | $32.72 \pm 15.13\%$ |
| + $\mathbf{v_z}$ **Fine-tuned FV** | $\mathbf{81.29} \pm 4.38\%$ | $\mathbf{89.98} \pm 1.80\%$ | $\mathbf{79.80} \pm 3.71\%$ | $\mathbf{60.79} \pm 13.39\%$ |

Table 7: Average accuracies and standard errors across the simple-task, complex-task, and SemEval datasets. Compared to the baseline and initial FV intervention, we found that the fine-tuned FV leads to the best accuracies for all prompt types. FV = function vector.

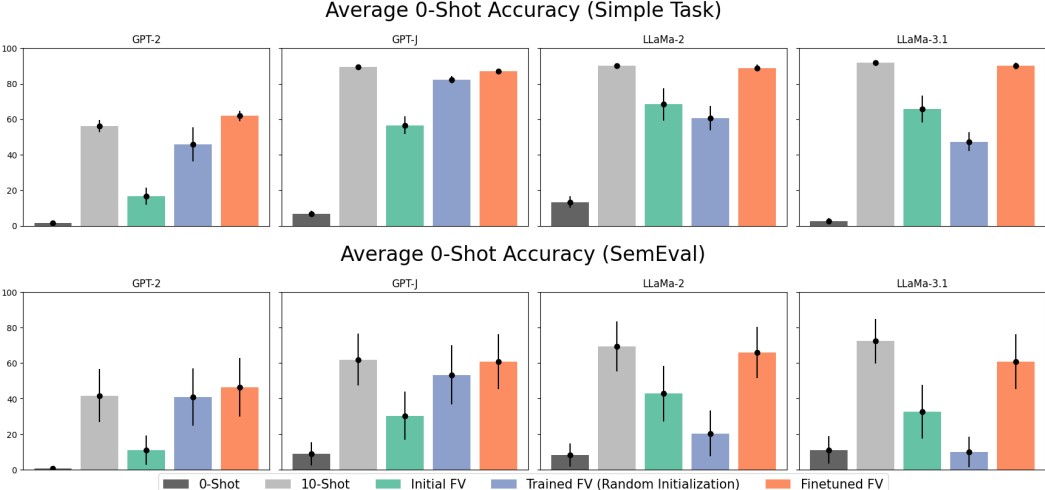

Figure 8: Average results for the Zero-shot evaluation of the simple-task dataset (top) and SemEval dataset (bottom). Compared to both the initial FVs (green) and trained FVs with random initialization (blue), applying the fine-tuned FV (orange) into the zero-shot task led to the best accuracies on average, similar to those for 10-shot learning (light grey). Additionally, the randomly initialized FVs for larger models require more training to maintain accuracies close to the fine-tuned FVs, emphasizing the need for initial FVs to fine-tune from. FV = function vector.

Alongside comparing the fine-tuned function vector with zero-shot learning and the initial function vector, we also compared it to 10-shot learning as well as a randomly initialized vector trained with

the same method. Since the initial FV is computed using 10-shot prompts, we tested if the FFV's performance in the 0-shot tasks match those of the 10-shot tasks where the relation can be easily determined. The randomly initialized vector was incorporated as another baseline to determine the necessity of using the initial FV for our finetuning method. From the results in Figure 8, we see that our FFV outperforms the randomly initialized FV while having competitive performances with 10-shot learning.

### A.3.1 Decoding Function Vectors

To examine whether relation words/phrases can be extracted from the FV, we construct a decoder that runs the input through layer 13 (the layer following FV intervention) and the last layer before decoding its output to the token distribution. While Todd et al. (2023) decoded the FV by itself, we added the attention head values from layer 12 for a blank prompt (i.e. "Q: A: "). As the FV is added to the hidden state of layer 12, there is additional information from the attention values that may lead to a more interpretable distribution when combined with the FV. As shown in Table 8, the resulting tokens either are within the relation's output space or tend to be representative of the relation itself. With the FFV, tokens that relate to the relation's abstract concept rather than its output space tend to be promoted. For example, under the Singular–Plural relation, FFV decoding retrieves the token "plural" whereas the top tokens from decoding the initial FV do not include any relevant relation terms.

| Task Name | Model | Decoded Tokens |
|---|---|---|
| Antonym | FV | ' unc', ' *opposite*', ' uncond', ' *opponent*', ' *oppos*' |
| | FFV | ' *opposite*', ' *oppos*', ' unc', ' *opponent*', ' vice' |
| capitalize_first_letter | FV | ' *acronym*', ' *abbre*', ' *initials*', ' *abbrevi*', ' adjective' |
| | FFV | ' *acronym*', ' *initials*', ' *abbre*', ' *abbrevi*', ' acron' |
| Singular-Plural | FV | ' Bs', ' balls', ' VE', ' rifles', ' frogs' |
| | FFV | ' Bs', ' *plural*', ' bags', ' rifles', ' warehouses' |

Table 8: Top-5 tokens decoded from the FV and fine-tuned FV, in the order of decreasing probability. Relation-relevant words are listed in italic. The fine-tuned FV appears to decode to more relation-relevant tokens compared to the FV. For example, the capitalize_first_letter FFV decodes to words associated with capitalizing the first letter, while 'plural' was one of the top-scoring tokens for the Singular-Plural FFV. FV = function vector, FFV = fine-tuned function vector.

### A.4 Composite Function Vector Evaluation

### A.4.1 Affine Transformation

The CFV's weights determine the relevance of each relation for a source analog. Such a representation can be translated into one that aligns closer to an analogy-focused latent space. While directly using the posterior distribution as the CFV's weights led to better accuracies than the baseline LLMs, applying the affine transformation $g(x) = Ax + b$ amplified the improvements. This is especially the case for the near-analogy problems in Green's dataset (+5.1% for GPT-2, +5% for GPT-J, +1.5% for LLaMa-2, +7.0% for LLaMa-3.1). Results across all four models are provided in Tables 9 and 10.

### A.4.2 CFV Results for BATS Dataset

BATS is a large-scale benchmark for evaluating analogy tasks, consisting of 2,000 analogy problems based on 40 relations grouped into two syntactic relation and semantic relation categories. Most BATS analogies are near-analogy problems characterized by well-defined syntactic or semantic relations instantiated within domains — for example, regular plurals (*student:students*), participle–past (*following:follows*), member–group (*player:team*), and part–whole (*wheel:car*). Consequently, model performance on BATS closely parallels results on near-analogy problems in Green's dataset. As shown in Figure 9, GPT-J with composite function vectors exhibited comparable performance, with only small improvements (e.g., a 3% increase for inflectional morphology, no change

| Ex. (*answer : riddle*) | Green's Analogy | | SAT Problems |
| | Near-Analogy (*solution : problem*) | Far-Analogy (*keys : locks*) | |
| --- | --- | --- | --- |
| GPT-2 | 50.0% | 22.5% | 19.8% |
| + IFV-CFV without affine | $48.0 \pm 1.0\%$ | $29.0 \pm 3.4\%$ | $22.4 \pm 1.3\%$ |
| + $g()$ affine | $47.5 \pm 3.4\%$ | $41.0 \pm 7.4\%$ | $23.2 \pm 3.0\%$ |
| + $\mathbf{v_C}$ without affine | $53.4 \pm 5.5\%$ | $38.5 \pm 1.2\%$ | $26.0 \pm 3.5\%$ |
| + $g()$ affine | $\mathbf{58.5} \pm 11.6\%$ | $\mathbf{44.5} \pm 8.7\%$ | $\mathbf{28.6} \pm 8.3\%$ |
| GPT-J | 75.0% | 42.5% | 33.4% |
| + IFV-CFV without affine | $69.5 \pm 1.0\%$ | $50.0 \pm 0.0\%$ | $34.8 \pm 1.1\%$ |
| + $g()$ affine | $69.5 \pm 5.7\%$ | $50.5 \pm 2.4\%$ | $36.7 \pm 4.5\%$ |
| + $\mathbf{v_C}$ without affine | $74.0 \pm 2.0\%$ | $56.5 \pm 4.2\%$ | $44.4 \pm 2.8\%$ |
| + $g()$ affine | $\mathbf{78.0} \pm 3.0\%$ | $\mathbf{61.0} \pm 5.4\%$ | $\mathbf{49.1} \pm 6.2\%$ |
| LLaMa-2 | 77.5% | 60.0% | 51.9% |
| + IFV-CFV without affine | $74.0 \pm 1.2\%$ | $67.0 \pm 5.4\%$ | $46.9 \pm 2.7\%$ |
| + $g()$ affine | $75.0 \pm 2.4\%$ | $69.0 \pm 7.9\%$ | $52.2 \pm 6.3\%$ |
| + $\mathbf{v_C}$ without affine | $73.0 \pm 5.5\%$ | $75.0 \pm 2.0\%$ | $55.0 \pm 5.2\%$ |
| + $g()$ affine | $74.5 \pm 2.5\%$ | $74.5 \pm 11.5\%$ | $\mathbf{56.3} \pm 9.2\%$ |
| LLaMa-3.1 | 87.5% | 57.5% | 52.7% |
| + IFV-CFV without affine | $82.5 \pm 2.4\%$ | $62.0 \pm 3.0\%$ | $48.9 \pm 6.5\%$ |
| + $g()$ affine | $82.0 \pm 3.4\%$ | $56.0 \pm 8.1\%$ | $50.1 \pm 7.0\%$ |
| + $\mathbf{v_C}$ without affine | $78.0 \pm 5.5\%$ | $72.0 \pm 12.6\%$ | $55.6 \pm 10.7\%$ |
| + $g()$ affine | $85.0 \pm 2.0\%$ | $\mathbf{74.0} \pm 10.1\%$ | $\mathbf{58.8} \pm 9.3\%$ |

Table 9: Average accuracies and standard errors for Green's and SAT datasets. As the entirety of each dataset is used for evaluation, the LLM's accuracy per word pair is consistently the same across seeds. $\mathbf{v_C}$ = composite function vector, IFV-CFV = composite function vector constructed from initial function vectors.

| | Morphology | | Semantics | |
| | Inflectional (*ability : abilities*) | Derivational (*able : unable*) | Encyclopedic (*cat : feline*) | Lexicographic (*bear : cub*) |
| --- | --- | --- | --- | --- |
| GPT-2 | $83.4 \pm 4.6\%$ | $59.0 \pm 4.4\%$ | $27.4 \pm 4.7\%$ | $32.2 \pm 5.7\%$ |
| + $\mathbf{v_C}$ without affine | $\mathbf{90.2} \pm 2.7\%$ | $\mathbf{66.4} \pm 5.0\%$ | $\mathbf{37.4} \pm 6.5\%$ | $\mathbf{45.4} \pm 6.3\%$ |
| + $g()$ affine | $83.6 \pm 5.1\%$ | $59.8 \pm 5.8\%$ | $33.5 \pm 6.1\%$ | $42.7 \pm 6.8\%$ |
| GPT-J | $95.3 \pm 3.2\%$ | $80.0 \pm 4.3\%$ | $65.2 \pm 6.3\%$ | $51.3 \pm 5.5\%$ |
| + $\mathbf{v_C}$ without affine | $\mathbf{99.5} \pm 0.6\%$ | $80.0 \pm 4.1\%$ | $62.9 \pm 6.4\%$ | $\mathbf{57.4} \pm 6.8\%$ |
| + $g()$ affine | $98.2 \pm 1.3\%$ | $\mathbf{80.4} \pm 5.2\%$ | $\mathbf{65.4} \pm 7.0\%$ | $54.5 \pm 6.0\%$ |
| LLaMa-2 | $98.0 \pm 1.7\%$ | $91.8 \pm 3.7\%$ | $78.6 \pm 4.6\%$ | $62.0 \pm 6.5\%$ |
| + $\mathbf{v_C}$ without affine | $98.2 \pm 1.5\%$ | $82.7 \pm 4.9\%$ | $70.5 \pm 5.4\%$ | $60.3 \pm 5.9\%$ |
| + $g()$ affine | $\mathbf{98.6} \pm 1.7\%$ | $88.6 \pm 4.9\%$ | $75.5 \pm 5.5\%$ | $58.6 \pm 6.6\%$ |
| LLaMa-3.1 | $99.4 \pm 0.3\%$ | $95.7 \pm 2.5\%$ | $87.4 \pm 4.5\%$ | $67.0 \pm 6.7\%$ |
| + $\mathbf{v_C}$ without affine | $\mathbf{100.0} \pm 0.0\%$ | $88.4 \pm 3.8\%$ | $77.9 \pm 5.5\%$ | $67.0 \pm 5.9\%$ |
| + $g()$ affine | $99.4 \pm 0.5\%$ | $87.0 \pm 6.8\%$ | $82.4 \pm 5.4\%$ | $64.9 \pm 6.7\%$ |

Table 10: Accuracies and standard errors for each group in BATS. The improvements from CFV intervention (both with and without the affine transformation) are primarily within GPT-2 and GPT-J. CFV = composite function vector.

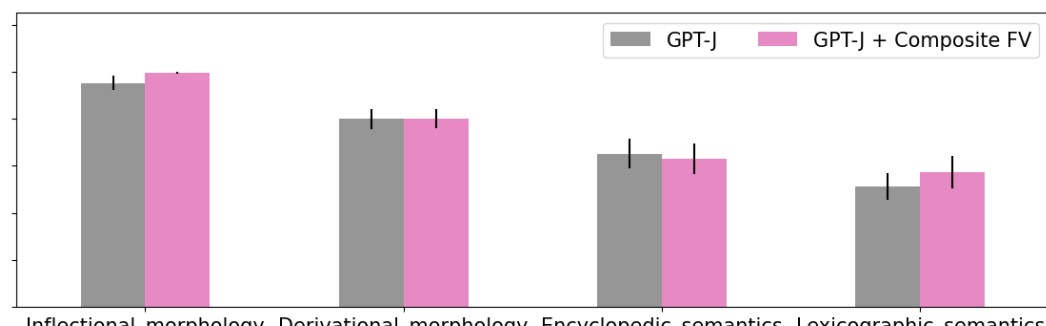

Figure 9: Top-5 prediction accuracy for the one-shot analogy problems in BATS for GPT-J.

for derivational morphology and encyclopedic relations, and a 3% increase for lexicographic relations). We further observe that CFVs boosted accuracy on semantic relations in small-scale LLMs such as GPT-2, yielding gains ranging from 5–10%. However, larger-scale LLaMA models exhibited slight performance drops after CFV injection, likely for the same reason as the decline in near-analogy accuracy on Green's dataset: these models rely more on pre-trained semantic associations than on relational knowledge to solve near-analogy problems. The results on BATS are provided in Table 10.

## A.5 RELATIONAL SIMILARITY

When examining the relational similarity across other models, we observe similar patterns for the activations under FV and FFV intervention. With FV intervention on larger-scale models like LLaMa-2 and 3.1, the activations for word pairs within the same relation type tend to exhibit higher cosine similarity, but this is attenuated with FFV intervention. Results for the other three models are in Figure 10.

| | Exp. 1 | | Exp. 2 | |
|---|---|---|---|---|
| | $r$ | 95% CI | $r$ | 95% CI |
| GPT-2 | .574 | [.52, .62] | .584 | [.53, .63] |
| + $v_t$ Initial FV | .778 | [.75, .80] | .730 | [.69, .76] |
| + $v_z$ **Fine-tuned FV** | **.807** | [.78, .83] | **.822** | [.80, .84] |
| GPT-J | .423 | [.36, .48] | .611 | [.56, .65] |
| + $v_t$ Initial FV | .779 | [.75, .81] | .668 | [.63, .71] |
| + $v_z$ **Fine-tuned FV** | **.788** | [.76, .81] | **.831** | [.81, .85] |
| LLaMa-2 | .693 | [.65, .73] | .730 | [.69, .76] |
| + $v_t$ Initial FV | .816 | [.80, .84] | .658 | [.62, .70] |
| + $v_z$ **Fine-tuned FV** | **.846** | [.82, .87] | **.747** | [.71, .78] |
| LLaMa-3.1 | .670 | [.63, .71] | .620 | [.57, .66] |
| + $v_t$ Initial FV | .811 | [.78, .83] | .645 | [.60, .69] |
| + $v_z$ **Fine-tuned FV** | **.864** | [.84, .88] | **.749** | [.72, .78] |

Table 11: Pearson correlations between model-predicted relational dissimilarity and human judgments. The FFV method consistently accounts for human judgments of relational similarity the best. FV = function vector, FFV = fine-tuned function vector, CFV = composite function vector.

## A.6 T-SNE

To examine the similarity across the FFVs themselves, we generate a t-SNE plot to visualize their clusters. As shown in Figure 11, there is clear clustering that separates syntactic relations from

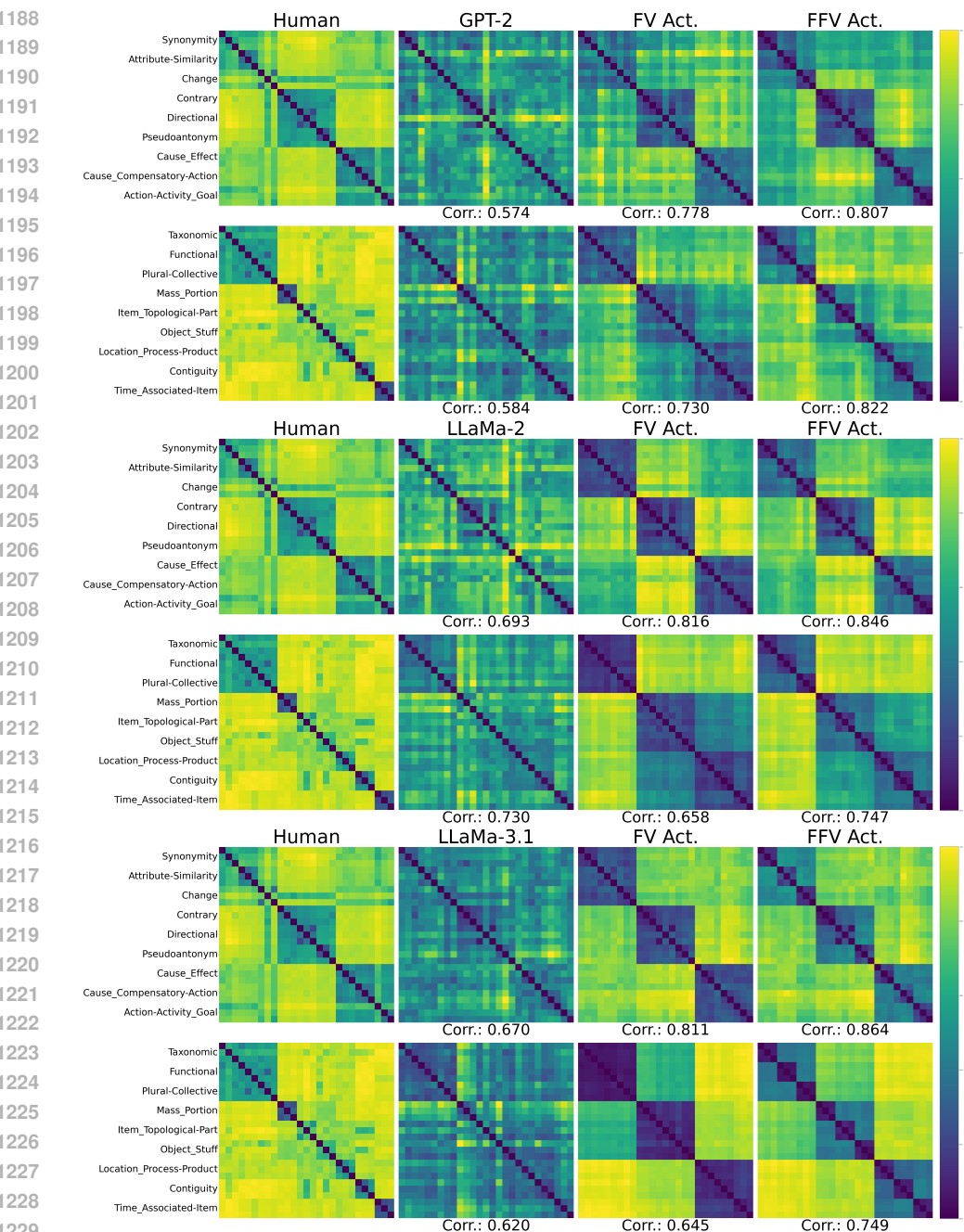

Figure 10: Relational dissimilarity matrices of human judgments and predictions generated by GPT-2 (top), LLaMa-2 (middle) and LLaMa-3.1 (bottom), both alone and with the FVs. In each panel, the top row represents Exp. 1 in Ichien et al. (2022), while the bottom row represents Exp. 2 with word pairs. The two experiments used different sets of relations for measuring human relation similarity judgments. While the activations of the initial FV-injected LLMs are tightly grouped by relation type, the FFV-injected LLMs account for dissimilarities across relations as shown in the similarity matrices. In general, the LLMs' activations tend to align closer to the human judgments with FV intervention, as evidenced by the structure of their similarity matrices and Pearson correlation. FV = function vector, FFV = fine-tuned Function Vector.

semantic relations. Specifically, we found that the FFVs computed from syntactic relations (from Google and MSR datasets) tend to cluster into one group, while those derived from semantic rela-

tions cluster according to their general relation type. This result provides additional evidence that FFVs capture fundamental structures of relational knowledge.

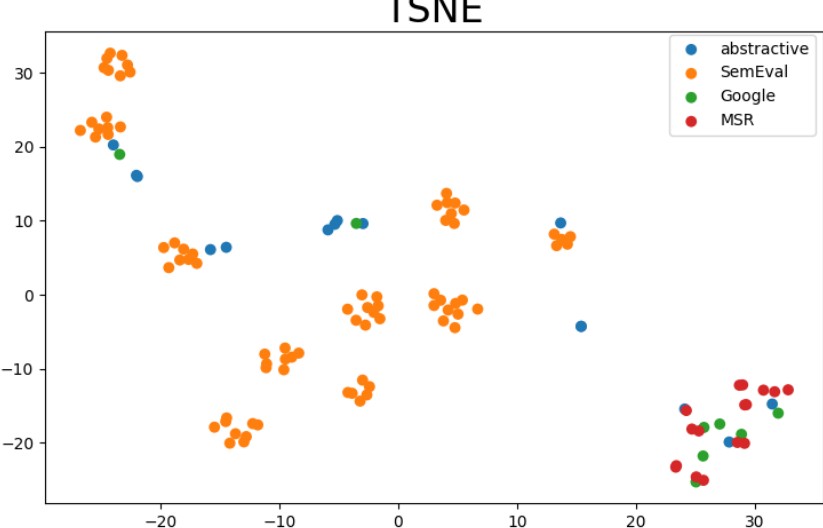

Figure 11: t-SNE of all FFVs computed with GPT-J. We found that the FFVs of syntactic relations (Google and MSR datasets) tend to be clustered together at the bottom right corner, while those of semantic relations (abstractive dataset from Todd's study and SemEval datasets) are clustered in the top left region. FFV = fine-tuned function vector.

## B  USE OF LARGE LANGUAGE MODELS

Aside from the study's research being focused on LLMs, ChatGPT was used to proofread a few paragraphs for more cohesive explanations. Otherwise, LLMs were not used for research ideation, data analysis, retrieval discovery, or any other critical contributions. The authors take full responsibility for the final content, including research ideas and results.

