# OpenReview forum: "Relational Knowledge Distillation Using Finetuned Function Vectors"
_ICLR.cc/2026/Conference — Submitted to ICLR 2026_

### Official Review · Reviewer_Qe3R · 2025-10-19

**Soundness:** 1
**Presentation:** 2
**Contribution:** 2
**Rating:** 2
**Confidence:** 4

**Summary:**

The paper introduces a simple approach to finetuning existing function vectors (FV) for relation tasks, known as finetuned function vectors (FFV). The approaches requires few samples and improves over several tasks that FVs are evaluated on. Furthermore, they also contribute an approach to combine FFVs into composite function vectors (CFVs) that captures analogical knowledge over multiple types of FFVs. They apply their approach to evaluation task cross-domain human similarity judgments.

**Strengths:**

1. The authors show that their FFV approach shows consistent increases in performance over the vanilla FV approach.
2. The paper corroborates findings from [1] showing random initialization for training function vectors has weaker performance than initialization via CMA (section 3.2 of [1]).
3. The application of the CFV is interesting as a direction to study human similarity judgments.

[1] Todd, Eric, et al. "Function Vectors in Large Language Models." *ICLR* (2024).

**Weaknesses:**

1. The work is initially motivated from the point of view of efficiency (lines 40-44), but no citations are introduced (or preliminary experiments to support the hypothesis) in lines 42-44 when referencing these factors -- line 43 "Many factors contribute to the inefficiency of the AI models in solving reasoning problems, but we suspect that one major issue is the lack of explicit representations for relations and the inability to accumulate the relational knowledge during learning". Efficiencies needs to be defined empirically and given a formal definition (is it efficiency via model throughput, efficiency via tokens generated, or others?) since the crux of the argument is related to efficiency.
2. It's unclear from the beginning of the paper what exactly is a relation task, even though the notation becomes slightly more clear towards the end of the paper. A more formal definition (even fairly brief) would be helpful as a preliminary. Many of the existing FV tasks are in some sense "relations" (antonyms, synonyms, etc.) and so the distinction is not clear, because the same "relations task" term is used later to refer to both the antonyms/synonyms task and the analogies task.
3. From the introduction, it is unclear why one would need to "improve" the performance of existing FV because the task (analogy task) is not introduced early enough in the paper in the motivation. It would be tremendously helpful to explain why existing FVs are not adequate enough to support these analogies (for instance, through some error analysis of FVs on these existing analogical tasks) and then show that additional training to FVs are required, because in some sense, showing that FFVs are stronger than FVs is obvious since finetuning will always improve over inference-only approaches.
4. In general, writing needs to be clearer. For example, in the experimental setting, there are many datasets, but it is relatively difficult to parse the particular categories of tasks used to train each and every FFV and to keep track of which ones are used. It would be nice to include a clearer breakdown in that section which clearly denotes what relations are being used, or even picking a subset of FVs to evaluate and putting the rest in the appendix (like in Todd et al.).
5. It would be stronger to motivate the work by introducing Ichien et al. 2022 earlier on, since it seems that the human similarity judgments are the core evaluation application for the work and it seems to be very interesting to study human cognitive behavior using FFVs/CFVs. However, the reader gets to the application far too late to realize this is a motivation, especially since it is mostly mentioned in the conclusion as a takeaway rather than core investigative application.
6. Some repetitive writing - line 403-410 is repeated in the experiments section lines 266-273.
7. Broken link on line 618-619 for the table.

Based on the existing motivation, much of the paper may need to be rewritten which makes me inclined to give a 2, even though the experiments are reasonable.

**Questions:**

1. What "relation" tasks are finetuned and used when determining the weighted combination for CFVs? Although a table exists in the appendix, it should be made clear in the main body of the paper since it is difficult to understand the CFV setup (what combinations are used in the CFV for inference).
2. Analogously with the previous question, how does the CFV learn to choose the specific FFVs to focus on? Is this determined on-the-fly with the weighted combination?
3. What is the performance of the CFVs on the same simple tasks on Table 1? I would be curious to see whether across the tasks, the behavior of the CFV is consistent with an individual FFV, and if the CFV captures the weighted combination correctly.
4. In the evaluation (Zero-shot evaluation), why is the antonym FV chosen and not other FVs? Why not show the result of selecting/running other FVs as baselines, since there could be in theory up to N number of baselines, where N is the individual FV task chosen/used?
5. Why was layer 13 chosen in Section 4.3 (line 428-429)?

---

> ### Author Response · Authors · 2025-11-20
>
> We thank Reviewer Qe3R for the thoughtful and constructive feedback. We have revised the paper substantially in response and address each point below.
>
> **Weaknesses 1-3 & 5:**
>
> We have substantially revised the introduction to strengthen the overall motivation of the work. In particular, we now frame it around the role of analogies and integrate human similarity judgments as a core motivation rather than a concluding remark. These revisions now establish a more coherent narrative around why improved relational representations are needed and how FFVs and CFVs address this gap.
>
> **Weakness 4:**
>
> Section 4.1 has been revised to clearly distinguish the datasets used for FFV training, the datasets used for evaluation, and those used when constructing CFVs. We include more explicit descriptions of each dataset, making the experimental setup easier to follow.
>
> **Weakness 6:**
>
> The repeated paragraph has been removed. The description of BATS in the Composite FV section (lines 318-331) has been revised to incorporate the relevant information without redundancy.
>
> **Weakness 7:**
>
> The broken link has been corrected - it now refers to Table 6 in the revised paper.
>
> **Question 1:**
>
> The relation tasks used to construct the CFV are now defined in the Composite FVs paragraph in Section 4.1 (Datasets). The revised paper clarifies exactly which FFVs contribute to CFV construction and how the relation pool is defined.
>
> **Question 2:**
>
> The CFV weights are computed dynamically by applying each FFV to the source analogy pair and obtaining a probability distribution over the relations from their resulting scores. Section 3.2 and Equation 2 in the revised paper provide a clearer and more direct explanation of this process.
>
> **Question 3:**
>
> The simple tasks in Table 1 rely on zero-shot and few-shot prompting, which is not directly applicable for CFVs as CFV construction requires a source analogy pair to compute relation probabilities from. For this reason, CFVs cannot be evaluated on the tasks in Table 1.
>
> **Question 4:**
>
> Zero-shot evaluations were conducted for all FVs across both the Simple-task and SemEval datasets. The antonym FV was presented in the text as a representative example, but the full set of results includes all FVs stated previously.
>
> **Question 5:**
>
> Layer 13 was chosen because our FV intervention occurs at Layer 12; examining attention activations in the following layer allows us to observe how the injected relational information starts to propagate through the model.

---

> ### Comment · Reviewer_Qe3R · 2025-11-26
>
> Thanks for addressing my concerns. I will maintain my score because I do not think that the paper in its current state is fit for publication. In particular, I still have a few questions (and few new ones):
>
> **Weaknesses 1-3 & 5**: The framing is more clear now. However, I am still not convinced that we should care about understanding analogical reasoning via LLMs. There's a few terms thrown around in the new introduction now like "relational knowledge during pretraining" and pretraining is not investigated here, for instance. How do the authors get from "pretraining" -> ICL -> FFVs?
>
> - I still do think the authors have not motivated the **why** about finetuning FVs; I echo the same sentiment as Reviewer P6Px about the why, because the finding it rather obvious here. I do not think that the authors have satisfyingly solved this tension. The motivation towards human studies is more clear here, but I'm still unconvinced that FFVs solve this problem. For instance, in **Figure 5**, it doesn't actually seem that FFVs are significantly better than using FVs (if anything, they look to be about the same performance for one correlation plot!). Especially since I don't actually know how the correlation scores are calculated on the basis of the existing caption, but it seems like you can shift around the particular rows to change the correlation scores so that it looks better for FFVs vs. FVs. Therefore, I am unconvinced that even finetuning FFVs may always be useful.
>
> - Again, it seems like this question has not been answered, only the paper introduction rewritten, which is not strong enough to support the claims made in the paper: **...(for instance, through some error analysis of FVs on these existing analogical tasks) and then show that additional training to FVs are required, because in some sense, showing that FFVs are stronger than FVs is obvious since finetuning will always improve over inference-only approaches.**
>
> **Q5**: do you have any particular citations for choosing the middle layer? Did you try other layers? What does it look like when you do so?
>
> **Additional Q1**: What does Section 4.2.3 add to the paper? It doesn't seem to be substantial, considering it is essentially the same takeaway and finding as one section in the FVs paper. FVs and FFVs will encode the behavior's abstract concept -- this is obvious. The fact that FFVs may encode the concept *more* is also obvious considering you are now training the FV.
>
> **Additional Q2**: Where is the baseline for the CFV where you do not finetune a FV and compose it, Secion 4.3? The only one that is tested is what I presume is Composite + FFV, which I assume notated incorrectly as Composite + FV.
>
> **Additional Q3**: How is the affine mapping for the CFV trained? This part is unclear from Section 3.2.

---

> > ### Author Response · Authors · 2025-12-01
> >
> > We thank Qe3R for the follow-up comments and constructive feedback.
> >
> > **Weakness 1**
> >
> > The pretrained knowledge serves as the foundation for LLMs to refer to when processing any new information - for in-context learning, the LLMs would leverage this knowledge to respond to the prompt accordingly. Because of this mechanism, we believe that the LLM’s relational knowledge can be distilled into a more interpretable representation.
> >
> > **Weakness 2**
> >
> > For Human Experiment 1 (top row of Figure 5), we agree that the improvement from the base FVs to the FFVs is modest (0.779 vs. 0.788). However, Human Experiment 2 (bottom row), which uses a different set of relations and word pairs, shows a substantially larger gain (0.668 vs. 0.831). We have included these detailed results into the main text (Lines 412-420) and report corresponding results for GPT-2, LLaMA-2, and LLaMA-3.1 in the appendix (Figure 10, Table 11). Across the four models tested (GPT-2, GPT-J, LLaMA-2, LLaMA-3), adopting FFVs rather than the base FVs aligns closer to human similarity judgments by 3-5% in Experiment 1 and 8.9-16.3% in Experiment 2.
> >
> > We also want to emphasize that FFVs were fine-tuned using ~20 word pairs per relation only for the relation-completion task; the enhanced alignment with human similarity judgments is not directly optimized during fine-tuning. While some improvement may be expected, the magnitude of alignment, especially in Experiment 2, was not anticipated.
> >
> > **Weakness 3**
> >
> > To further compare the initial FVs and the FFVs, thereby necessitating fine-tuning, we conducted a systematic comparison in solving analogy problems. We found that adopting FFVs in the CFV rather than the base FVs increased accuracies by 2-18% for Green’s far-analogy problems and 6.7-10.1% for SAT analogy problems across the four models tested (GPT-2, GPT-J, LLaMa-2, LLaMa-3.1). We present the results in Table 9 of the revised paper.
> >
> > **Question 5**
> >
> > For FV intervention, we chose the middle layer as it consistently led to the highest accuracies across relations for the original FV intervention. Applying the FFV to earlier layers on inference also yields significant improvements like the initial FV, but the accuracies are more consistent until the later layers where accuracy drops. We discuss this in Figure 7 on page 17, which also visualizes the layer sweeps.
> >
> > **Additional Question 1**
> >
> > Our goal in decoding the FVs is to inspect what types of lexical evidence the vectors prioritize, as there is more interpretable information that can be extracted from the FFVs vs. the initial FVs. . Unlike the original FVs where Todd et al. found that the tokens usually resemble the relation’s outputs [3], FFVs tend to promote tokens that reflect the abstract concepts of their relations instead. This provides a useful diagnostic for how they encode interpretable relational structure.
> >
> > **Additional Question 2**
> >
> > The baselines for the CFV without the affine transformation are in Table 9 of the revised paper. We also want to clarify the notation in Figure 6: “LLM + Composite FV” refers to the LLM with the composite function vector injected to it.
> >
> > We found that the affine transformation is particularly helpful for smaller models (e.g., GPT-2), while providing more limited gains for larger models (e.g., LLaMA-3.1). For GPT-2, the baseline model achieved 22.5% accuracy on Green’s far-analogy problems and 19.8% on SAT analogies. Using the CFV without the affine transformation improved these to 38.5% and 31% respectively. Adding the affine transformation yielded further improvements, with accuracies of 43% on Green’s problems and 32.7% on SAT.
> > For LLaMA-3.1, the baseline model reached 57% accuracy on Green’s analogies and 52.7% on SAT problems. The CFV alone substantially improved performance to 72% and 61.3%. The affine transformation offered only modest additional benefits, raising Green’s accuracy to 74% and leaving SAT performance unchanged at 61.3%.
> >
> > **Additional Question 3**
> >
> > The affine transformation is trained using backpropagation using the same method as for the FFVs - while we bring this up in Appendix Section A.4.1, we have also clarified this in lines 261-263 and 276-277 of the main text.
> >
> > [1] Todd, Eric, et al. "Function vectors in large language models." (2023)

---

### Official Review · Reviewer_oDLE · 2025-10-29

**Soundness:** 2
**Presentation:** 2
**Contribution:** 2
**Rating:** 4
**Confidence:** 3

**Summary:**

The paper proposes finetuned function vectors (FFVs) that adapt the “function vector” representation through lightweight gradient updates while keeping LLM weights frozen. It further introduces composite function vectors (CFVs) that are weighted sums of FFVs computed from a source pair to steer models on one-shot analogies. Empirically, FFVs substantially outperform un-tuned FVs on zero-shot relation completion across many base LLMs, and CFVs notably improve cross-domain analogies; the authors also report stronger alignment with human relational similarity judgments.

**Strengths:**

1. Originality. Extends function vectors (compact task representations identified via causal mediation) with a practical fine-tuning scheme and a linear composition mechanism for analogical transfer—bridging activation steering and relational reasoning.
2. Quality. Clear problem setup; simple objective for FFVs; broad evaluation over multiple LLMs and datasets with consistent zero-shot gains and credible human-similarity correlations.
3. Clarity. The pipeline and training/inference diagrams are easy to follow; datasets and tasks are well cataloged.
4. Significance. Demonstrates controllable, inference-time manipulation of relational knowledge with measurable benefits on far analogies and human-like relational structure.

**Weaknesses:**

1. The work focuses on word-pair relations and four-term analogies; broader compositional or sentential relations remain open. Testing FFVs/CFVs on sentence-level or narrative analogies would strengthen external validity.
2. CFV weights are derived over a curated set of relations and fitted via an affine map; the sensitivity of performance to pool content/size and calibration is unclear. Please include pool ablations and out-of-distribution relations.
3. Ablations on injection and capacity. FFVs are injected at a fixed layer; maybe the authors should provide layer sweeps, magnitude schedules, and head-wise sparsity to clarify where steering works.
4. Writing/format nits. Minor issues reduce polish (e.g., duplicated “the this study’s” phrasing in limitations; a “Table ?? ” placeholder; equation formatting around the affine map). A careful pass would help.
5. The main limitations are scope (word-pair analogies), some sensitivity/design choices around CFVs, and a need for deeper ablations and polish.

**Questions:**

1. Can you learn a confidence or gating signal to decide when CFV steering helps (e.g., far vs. near analogies), avoiding regressions on easy cases?
2. How stable are CFV weights under perturbations to the relation pool (drop/merge relations; add noisy ones)? Provide sensitivity and sparsity analyses.
3. Could FFVs extend to sentence-level relations or graph-structured analogies (e.g., cause-effect narratives)? Any early evidence?
4. The correlations to human judgments are strong; can you localize which heads/layers contribute most to this alignment? Ablations or causal tracing would be illuminating.
5. Please position FFVs/CFVs more explicitly against activation-addition and related steering/editing methods.

**Details Of Ethics Concerns:**

No ethics concern.

---

> ### Author Response · Authors · 2025-11-20
>
> We thank Reviewer oDLE for the thoughtful and constructive feedback. We have revised the paper substantially in response and address each point below.
>
> **Weakness 1:**
>
> We acknowledge that the present work is limited to word-pair relations and four-term analogies. However, as noted in the revision, even large-scale LLMs perform poorly on challenging analogy problems, such as Green’s far-analogy tasks and SAT analogies. We show that both FFVs and CFVs consistently improve performance on models of varying sizes. Extending FFVs to sentence-level analogies is a natural next step - since FFVs already operate on pairwise relational structures, they could be applied to reasoning tasks with pairwise relations between entities. We outline this direction more explicitly in the discussion section as a promising avenue for future work.
>
> **Weakness 2:**
>
> Out-of-distribution relations are already represented in Green’s far analogies and SAT problems, and performance on these tasks remains stable even when the Google and MSR datasets are removed from the training pool. In contrast, performance on BATS decreases under this ablation, likely because several BATS relations are syntactic (e.g., present-past tense from inflectional morphology) that are no longer represented after these removals.
>
> **Weakness 3:**
>
> We include a layer sweep for FFV intervention in Supplemental Figure 7. As with the original FVs, performance gains are most prominent in earlier layers, but the improvements are more stable across relations. The revised text (lines 828-836) elaborates on injection depth, magnitude, and consistency.
>
> **Question 1:**
> A confidence or gating signal is a useful suggestion. In our current evaluation sets (Green’s far analogies and SAT analogies) easy cases are relatively rare, so we would not observe many situations where CFV intervention would produce regressions.
>
> **Question 2:**
>
> We appreciate the suggestion for sensitivity analyses. However, because our relation pool has meaningful structure (syntactic vs. semantic relations, distinct relational categories), random deletions may not yield interpretable results. The ablation study would need to be structured carefully with controlled pool manipulations.
>
> **Question 3:**
>
> We believe there is potential in FFVs to extend to sentence-level and graph-structured analogies. Preliminary results on problem-solving tasks (e.g., the commonsense_qa dataset) suggest that FFVs can support more complex reasoning beyond word pairs. One feasible extension is to apply CFVs to entity pairs with stories as context, supporting analogical reasoning at sentence level. We highlight this direction in the revised paper.
>
> **Question 4:**
>
> Localizing which attention heads or layers contribute most to human-aligned relational structure would be informative. However, due to the costs of collecting such data our human similarity dataset includes only 54 word pairs, leading to challenges with reliable identification of attention heads or circuits in LLMs.
>
> **Question 5:**
>
> We acknowledge the lack of comparisons with other steering methods in the discussion. However, we also emphasize that our method learns from only ~20 word pairs per relation, offering a compact and data-efficient alternative to existing steering techniques.

---

### Official Review · Reviewer_177W · 2025-10-30

**Soundness:** 3
**Presentation:** 2
**Contribution:** 2
**Rating:** 2
**Confidence:** 4

**Summary:**

This paper studies how to distill relational knowledge in LLMs into steering vectors. They show that finetuning previously found function vectors (FVs) improves performance compared to the original approach on analogy-style problems. They evaluate on many datasets of analogy problems, and compare representations to human judgments, showing that fine-tuning the representations gives the best fit.

**Strengths:**

- The fine-tuned function vectors (FFVs) are evaluated on a wider array of analogy problems compared to previous work, including analogy tasks like SemEval and BATS.
- The finetuning/composite function vector approach seems to improve far-analogy performance consistently across the board.

**Weaknesses:**

- The main contribution seems to be showing that by fine-tuning function vectors, you can get better performance than the original formulation, and this is tested on a larger set analogy-style tasks. While this is nice to see, I’m not sure how motivating/exciting of a result it is. This is for two reasons: (1) fine-tuning usually improves performance on downstream tasks, and (2) the analogy tasks chosen are rather simple.
    - Would this method work for more complex reasoning tasks such as multiple choice question answering, arithmetic, or ARC? These were recently proposed as sample subtasks of the “mechanistic interpretability benchmark” ([MIB](https://openreview.net/forum?id=sSrOwve6vb)) for instance.
    - Another way to strengthen the argument that this function vector finetuning (or composite) approach is a competitive alternative “steering method” is to evaluate on [AxBench](https://openreview.net/forum?id=K2CckZjNy0).
- There is little comparison to other methods/benchmarks for steering model behavior. If steering is the primary application of the method, it would be nice to see baselines like prompting, task-specific fine-tuning (of the model parameters), and others compared to FFVs in the main body of the paper. Some of this comparison to few-shot prompting is in the appendix (Tables 5, 6 and Figure 7), and briefly mentioned in the paper (eg. line 414) – where steering actually does worse than just prompting in some cases. I think including some of this information in the plots in Figure 1/Table 4 would help better contextualize the results (e.g. showing the results compared with prompting was helpful to see in Figure 7 to know how much of raw performance your FFVs are recovering).
- The composite function vector results are somewhat mixed. The results for the BATS dataset suggest that the CFV does not do much for these cases, and the only real “win” here is the far analogy case in the Green dataset. Can you explain the difference between these cases in more detail? I didn’t quite understand why these analogy sets are harder/easier.

**Questions:**

My main concerns I would like to see addressed are listed above, but I have a few additional questions would help clarify some confusions I had regarding them.

- The explanation and purpose of the composite function vector was a bit unclear to me. Is it just a weighted sum of all previously found function vectors that’s trying to capture “untrained” relations? What is meant by “projecting probabilities into analogy space” (line 215)?

- How does your fine-tuned function vectors approach compare to other external training methods that could induce task behavior such as Gist tokens ([Mu et al.](https://openreview.net/forum?id=2DtxPCL3T5)), soft prompts, codebooks [Shao et al.](https://openreview.net/forum?id=6axIMJA7ME3) LoRAs, etc. It seems like these approaches are similar in spirit to "learning a steering vector" for task behavior (and composition to compare to your CFV), but they weren't tested on the analogy style tasks you use.


- Minor Typos:
  - Line 234: “activities” -> activations?
  - Line 283: A right closing parenthesis is missing

---

> ### Author Response · Authors · 2025-11-20
>
> We thank Reviewer 177W for the detailed and constructive feedback. We have revised the paper substantially in response and address each point below.
>
> **Weakness 1:**
>
> We agree that improved zero-shot performance after fine-tuning is not, in itself, unexpected. However, our contribution goes beyond demonstrating performance gains. We demonstrate that FFVs
> 1. exhibit stronger alignment with human relational representations, reflected in higher correlations with human relation-similarity judgments;
> 2. yield more interpretable decoding; and
> 3. generalize to challenging analogy problems involving relations not encountered during training.
>
> Together, these results show that FFVs support SAT-style analogical reasoning as well as semantically distant analogy problems, extending beyond straightforward performance gains.
>
> **Weakness 2:**
>
> We appreciate the reviewer’s suggestion to compare against more steering baselines. However, applying existing steering methods directly to analogy tasks is challenging - each analogy requires a different underlying relation, while many steering techniques are designed for single, fixed tasks. For this reason, our primary comparisons center on
> 1. the initial FV identified via causal mediation analysis, and
> 2. generalization on analogy tasks across a diverse set of relations.
>
> Moreover, our method operates in a small-sample training regime (only ~20 pairs per relation), whereas alternative steering approaches (i.e. few-shot prompting) often assume significantly more information.
>
> We also want to clarify the point mentioned by the reviewer that “steering sometimes performs worse than prompting”: in Figure 7, prompting uses a 10-shot in-context learning setup with 10 demonstration pairs, while FV injection is evaluated strictly in the zero-shot setting.
>
> **Weakness 3:**
>
> Many relations in BATS overlap with those used during FFV training, so performance gains there largely reflect the composite function vector (CFV)’s ability to consolidate relations it has already learned. In contrast, the Green far-analogy dataset contains relations that are markedly out-of-distribution, making it a more direct test of whether CFVs can support genuine relational generalization. In the revision, we also include results on SAT analogy problems, which feature more advanced vocabulary and relation types not present during training, further evaluating generalization.
>
> **Question 1:**
>
> The reviewer's understanding of the CFV is correct: it is a weighted combination of previously learned FFVs intended to model relations that were not directly trained. We have expanded the motivation for this approach by drawing on relevant findings in psychology and neuroscience regarding analogies.
>
> **Question 2:**
>
> We acknowledge the lack of comparisons with other external training and steering methods in the discussion. However, we also emphasize that our approach is designed for - and empirically succeeds in - the small-sample regime stated in our response to Weakness 2 (i.e. 20 pairs/relation), which differentiates it from methods that depend on larger-scale task-specific optimization.
>
> Finally, we have corrected the typos noted by the reviewer.

---

> ### Comment · Reviewer_177W · 2025-11-26
>
> Thanks for the reply and helpful clarifications. I have read the updated manuscript, and I appreciated the update of the introduction, I think it better motivates why analogies are potentially more interesting to study than I initially gave them credit. I will update my score slightly based on the improvements made to the paper. However, I think there are still two outstanding concerns I have with the paper: limited comparison to other methods, and evaluation on more diverse forms of data beyond analogies.
>
> While the FFV does generalize to other analogy problems (and this is nice!), I think this is still a somewhat narrow test format/domain, and would've liked to see evaluating this process on other, more complex formats/domains.
>
> > " applying existing steering methods directly to analogy tasks is challenging"
> I appreciate the acknowledgment of the limited comparison with other baselines, but it makes the contribution harder to judge. Is it really true that other baselines do not work in a limited data regime? For instance, activation steering [Turner et al], or mass-mean probing [Marks et al] seem like they might be good candidates, but without comparison it is hard to really know.
>
> ___
> Turner et al [Steering Language Models With Activation Engineering](https://arxiv.org/abs/2308.10248)
>
> Marks et al [The Geometry of Truth: Emergent Linear Structure in Large Language Model Representations of True/False Datasets](https://openreview.net/forum?id=aajyHYjjsk)

---

> > ### Author Response · Authors · 2025-12-01
> >
> > We appreciate 177W for the score update and follow-up comments!
> >
> > **Concern 1: Evaluation on diverse data**
> >
> > For the FFVs, we have also observed some improvements in more complex formats like problem-solving tasks (i.e. CommonsenseQA, Sentiment Analysis datasets) with increases in accuracies from the initial FVs by 8.66-90.12% in GPT-J. WeGPT-J.  we report these results in Figure 4 (top right) of the revised paper.
> >
> > **Concern 2: Comparison to other methods**
> >
> > We thank the reviewer for the baseline suggestions. We agree that activation steering would be a useful baseline to compare our FFVs with. We have reported our accuracies for two steering vectors from ActAdd (one with only the relation’s activations $h_+$, and one with the counterbalancing approach $h_+ + h_-$) alongside mean-centering [1] in Section 4.2.1, Table 18 of the revised paper.
> >
> > While mass-mean probing is also a valid baseline for steering vectors, its evaluation criterion only measures the truthfulness of statements through binary classification (i.e. “The city of Tokyo is the capital of Japan” -> “True”) rather than its accuracy in predicting the correct answer from the query (i.e. “JapanTokyo” -> ?“Japan”). The probes were compared to logistic regression and steer the activations by transforming them to an individual score for binary True/False classification, rather than a distribution of scores that FVs use to determine the best token for completing a prompt. Adjusting the probes to align with our tasks (i.e. activations of “Japan -> Tokyo” vs. “America -> Tokyo”, obtaining score of the target token with softmax instead of sigmoid) led to poor accuracies (0-9%) in turn.
> >
> > [1] Jorgensen, Ole, et al. "Improving activation steering in language models with mean-centring." (2023).

---

### Official Review · Reviewer_P6Px · 2025-11-01

**Soundness:** 1
**Presentation:** 1
**Contribution:** 1
**Rating:** 2
**Confidence:** 4

**Summary:**

This paper proposes a approach to better capture the relational information in ICL by fine-tuning the Function Vector from [Todd et al, 2023](https://arxiv.org/pdf/2310.15213), which the authors dub as FFV (Fine-tuned Function Vector). The authors also show that a weighted combination of the FFVs can be used to capture the relational information from a 4-term analogical reasoning task. These composite function vectors can be used to even improve the LM performance on cross-domain analogy tasks.

**Strengths:**

I found the idea of composite function vectors quite interesting. It is not immediately obvious to me why how a weighted combination of FVs capturing other relations can help capture a new relation. But I find it quite fascinating that the authors found that it does in practice.

**Weaknesses:**

* Lack of theoretical justifications. The authors do provide some logical justifications for their approach, but I find them to be quite weak and unconvincing.
* The work do not introduce any solid novel ideas. It very much feels like an engineering effort to adapt FVs in the chosen analogy task setting. And often these design choices are not well justified.

**Questions:**

1. You finetune the FVs and get FFVs and show that they can achieve better performance on your shuffled and zero-shot evals. Isn't this kind of obvious? In my opinion, the most interesting claim from [Todd et al, 2023](https://arxiv.org/pdf/2310.15213) was that the LM has naturally developed specialized modules (in this cate attention heads) that capture such relational information. Of course you get better performance with finetuning. As long as the LM can perform the target task, and you are inserting your intervention (which is getting finetuned) early enough in the forward pass, you should be able to make the LM perform the task better. So, I don't find this contribution to be very significant. But I would love to hear the authors' thoughts on this.

    * (1a) Line 153-155: I am also very surprised that the authors used perplexity as their loss. I had the idea that cross-entropy usually considered standard for this. Perplexity is mainly used as an evaluation criterion. What was the justification for this choice?

2. Line 212-214: Not sure I understand that claim that you get "the posterior distribution". I am assuming $q$ is the query word, $r$ is the response, and $z$ is the relation or its function vector $v_z$. You get $[P(r | q, z_1), ..., P(r | q, z_n)]$ and apply softmax. And, then you say that you get $P(z | q, r)$? If my understanding is accurate the statement that you get a ""posterior distribution"" is not accurate, at least not in the Bayesian sense. You should probably say something like a proxy of P(z | q, r), which unit/elemental relations explain the target relation the most.

    * (2a) Immediately after these lines, you say that you need an affine transformation to project probabilities to the analogy. Your equation 4 shows that you get "posterior" *after* this transformation. What is the justification for this transformation? What is the affine transformation doing here?

3. As a justification for Composite Function Vectors, you just cite linear representation hypothesis and how FVs can serve has basis for constructing other relations. I find this justfication to be quite hand-wavy and simply not convincing.

     * (3a) You are making an assumption about compositionality: why should we expect a novel relation such as functionality of an apparatus can be composed as synonymy, antonymy ... ?
     * (3b) You take 100+ such "basis" relations. If you are claiming that you can compose any analogical relations that does that mean that these 100+ relations form a complete basis set?
     * (3c) One example of a far analogy you show is *blindness: sight :: poverty: wealth*. This is still an examply of antonym, which is one of your "basis" relations. I don't understand how compositionality is playing a role here.

4. I found several statements in the introduction to be quite confusing. I suggest the authors clarify these points.

     * (4a) For example, in lines 57 - 63, you draw analogies to how you aim to transform "implicit relational knowledge in LLMs into explicit knowledge that can be stored and manipulated to make inferences". What do you mean by "explicit knowledge"? Your FFVs? And, what is the "implicit" part here?
     * (4b) You draw some analogies with cognitive science about how humans acquire permanent knowledge. How is this relevant to your work? Your approach is still an intervention introduced at inference time.

###
Typo: Fix all the quotation marks.

---

> ### Author Response · Authors · 2025-11-20
>
> We thank Reviewer P6Px for the detailed and constructive feedback. We have revised the paper substantially in response and address each point below.
>
> **Weaknesses:**
>
> We revised most of the introduction to motivate the problem more clearly and highlight the role of analogy in our approach. We appreciate the reviewer’s suggestions, which helped us streamline the narrative and sharpen the motivation.
>
> **Question 1:**
>
> We agree that improved zero-shot performance after fine-tuning is not, in itself, unexpected. However, our contribution goes beyond demonstrating performance gains. We demonstrate that FFVs
> 1. exhibit stronger alignment with human relational representations, reflected in higher correlations with human relation-similarity judgments;
> 2. yield more interpretable decoding; and
> 3. generalize to challenging analogy problems involving relations not encountered during training.
>
> Together, these results show that FFVs support SAT-style analogical reasoning as well as semantically distant analogy problems, extending beyond straightforward performance gains.
>
> **Question 2:**
>
> We clarified this section in the revision. The affine transformation is introduced because several FFVs correspond to overlapping or closely related relations. For instance, an “opposite” relation appears in multiple datasets (simple-task, SemEval, MSR), each with different word pairs and slightly different relational nuances. Instead of manually selecting or pruning these basis relations, we learn an affine transformation $g(x) = Ax + b$ that integrates the relation probabilities in a data-driven manner. The transformation helps resolve redundancy and produces a coherent composite representation for one-shot analogy solving. This rationale is explained in lines 251-258.
>
> **Question 3:**
>
> The reviewer raises a valid concern regarding the assumption of compositionality. We now cite relevant work from psychology and neuroscience demonstrating the plausibility of relational composition (lines 219–223). Our results serve as a proof of concept that linear combinations of FFVs can address a wide range of analogy problems, including those involving semantically distant relations.
>
> Regarding the reviewer’s example - “blindness : sight :: poverty : wealth” - we agree that this is not a straightforward antonym. The antonym FFV is trained on pairs such as “hot : cold,” “love : hate,” and “long : short,” while the example represents a “lack of” relation. This distinction highlights how the learned representation space supports nuanced interactions among related but non-identical relations, which is central to analogical reasoning.
>
> **Question 4:**
>
> We have rewritten the introduction to emphasize the role of analogical reasoning in our work rather than the usage of implicit/explicit knowledge. This also helps better explain their relevance to the field of cognitive science: they provide motivation for representing relational knowledge in a stable, extractable form, even though our intervention is applied at inference time.

---

> > ### Comment · Reviewer_P6Px · 2025-11-21
> >
> > Thank you for your response.
> >
> > Q1:
> >
> > 1. I don't see significantly higher correlation with human judgments when compared with the base FVs. And, to repeat myself, finetuning helps the LM to better capture the task information, which might be the cause why this gives (sligthly) better correlation with human judgments. What is so surprising about this?
> >
> > 2. Why do I care about better decoding performance? There is something nice about the vectors encoding relational information can also verbalize that relation, but I don't understand this justification very well. Also, if decoding is so important, why not just use the `lm_head` row of the token corresponding to the relation (`" acronym"`) directly as a steering vector, instead of going through finetuning?
> >
> > 3. Where is that information? Don't you get a FFV per relation? What do you mean generalize to settings not seen during training?
> >
> > Q2: I packed a few different questions here. I think you are only answering one of them. On affine transformation, why can't you just filter out duplicate relations from different datasets? When you calculate $[P(r | q, z_1), ..., P(r | q, z_n)]$, and apply softmax, don't you already get "basis" FVs? Anyways, I still don't think the use of affine transformation is well justified.
> >
> > Q3. This is not what I was asking. I was asking if the authors have any idea how it makes sense theoretically and mechanistically.
> >
> > Q4. Can you clarify 4b here, please? In your response, please address the questions directly and/or summarize what changes you made to address these raised concerns.

---

> > > ### Author Response · Authors · 2025-12-01
> > >
> > > We thank Reviewer P6Px for the thoughtful follow-up and feedback. We address each point below.
> > >
> > > **Question 1**
> > > 1. In Human Experiment 1 (top row of Fig. 5), we agree that the gain from base FVs to FFVs is modest (0.779->0.788). However, Human Experiment 2 (bottom row), which uses different relations and word pairs, shows a much larger gain (0.668->0.831). These results are now in the main text (Lines 412-420) with GPT-2, LLaMA-2, and LLaMA-3.1's results in the appendix (Fig. 10, Table 9). Across all four models, FFVs align closer to human similarity judgments by 3-5% in Exp. 1 and 8.9-16.3% in Exp. 2.
> > >
> > > We want to emphasize that FFVs were fine-tuned on ~20 word pairs per relation only for relation completion; alignment with human judgments was not directly optimized. Thus, the magnitude of alignment, especially in Exp. 2, was not anticipated.
> > >
> > > We also compared CFVs derived from FFVs vs. base FVs on analogy solving. FFVs increased accuracy by 2-18% on Green’s far-analogies and 6.7-10.1% for SAT analogies across the four models (Table 9).
> > >
> > > 2. Our goal in decoding the FVs is to inspect what types of lexical evidence the vectors prioritize. FFVs tend to promote abstract relational terms - for example, decoding the Singular-Plural FFV retrieves “plural” which is absent when decoding its base FV. This offers insight into how they encode relational structure.
> > >
> > > Using the relation token as a steering vector through ActAdd only improved GPT-J from 5.1%->18.5% vs. the FFV at 75.2% on layer 15 (the best layer for mean-centering). We cover this in Sec. 4.2.1, Table 1.
> > > As Todd et al. note, FVs/FFVs contain critical information beyond the decoded token distribution; in Sec. 3.2 [1], the FVs' top 100 tokens yielded suboptimal accuracy; we expect the same for FFVs.
> > >
> > > 3. We apologize for the lack of clarity on our previous response. Although we train one FFV per relation, we form a CFV by taking weighted combinations of FFVs. This CFV encodes a new relational representation not explicitly trained, allowing the model to tackle analogy problems based on previously unseen relations. Many relations in Green’s dataset (e.g., answer : riddle :: key : ?) are absent from training, yet CFVs constructed from existing FFVs still support solving these problems, demonstrating generalization beyond trained relations.
> > >
> > > **Question 2**
> > > The affine transformation captures variation across overlapping relations that may subtly differ across datasets. (e.g. “opposite” as incompatibility in one dataset vs. contradiction in another). Removing overlapping relations could miss these distinctions - the affine transformation accounts for such shifts.
> > >
> > > The transformation's benefit is larger for smaller models (e.g., GPT-2) and more limited for larger models (e.g., LLaMA-3.1) as shown in Table 9 on page 210. For GPT-2, accuracies on Green's far-analogy problems were 22.5% (baseline), 38.5% (CFV), and 43% (CFV + affine); for SAT analogies: 19.8% (baseline), 31% (CFV), and 32.7% (CFV + affine).
> > > For LLaMA-3.1, the accuracies on Green’s analogies were 57% (baseline), 72% (CFV), and 74% (CFV + affine); for SAT analogies: 52.7% (baseline), 61.3% (CFV), and 61.3% (CFV + affine).
> > >
> > > **Question 3**
> > > - Theoretically, CFVs processing novel relations from a comprehensive set of basis relations makes sense because humans and LLMs generalize from prior knowledge to novel situations. LLMs often rely on analogical processes [2] rather than rule-based learning, using prior examples for pattern induction [3]. In turn, CFVs help reveal how LLMs use such analogical processes to generalize semantics.
> > > - Mechanistically, CFVs offer insight into the LLM’s reasoning processes by representing unseen relations as a weighted linear combination of basis relations; relevant relations are emphasized whereas irrelevant ones are down-weighted.
> > >
> > > **Question 4**
> > > We draw parallels to cognitive science as follows. Humans acquire relational knowledge gradually (e.g., children learning “opposites” at around age 5 [4]); similarly, pretrained LLMs slowly accumulate relational knowledge from large-scale training data.
> > > When solving tasks, humans retrieve relational knowledge into working memory; analogously, base FVs rely on in-context learning and causal mediation analysis to identify attention heads encoding relational information. We fine-tune these FVs with only ~20 word pairs per relation, akin to rapid refinement of relational concepts from a few examples in children.
> > > CFVs parallel working memory in human analogical reasoning: they combine relational representations to support flexible inference on new analogy problems.
> > >
> > > [1] Todd, Eric, et al. "Function vectors in large language models." (2024).
> > > [2] Hofmann, Valentin, et al. "Derivational morphology reveals analogical generalization in large language models." (2025).
> > > [3] Webb, Taylor, et al. "Emergent analogical reasoning in large language models." (2023).
> > > [4] Gentner, Dedre. "The development of relational category knowledge." (2005).

---

### Author Response · Authors · 2025-12-01
**Official Comment to Area Chair**

Since our original responses to the reviewers we have made the following major revisions to our paper:

- We have substantially revised the introduction to strengthen the overall motivation of the work. In particular, we now frame it around the role of analogies and position human similarity judgments as a core motivation rather than a concluding remark. These revisions now establish a more coherent narrative around why improved relational representations are needed and how FFVs and CFVs address this gap.
- We have also expanded our evaluation of FFVs to include complex-task relations from Todd et al. [1] (e.g., CommonsenseQA and Sentiment Analysis), as described in lines 298-305 of the revised paper. These experiments demonstrate that FFVs generalize to more complex formats and substantially outperform the initial FVs, yielding improvements of 8.66-90.12% on GPT-J. We report these results in Figure 4 (top right).
- We also evaluate FFVs alongside two activation steering methods: ActAdd [2] and mean-centered FVs [3]. For ActAdd, we used two steering vectors - one with only the relation’s activation vector $h_+$, and one with the counterbalancing approach $h_+ + h_-$. All methods were evaluated on the six representative relations in the simple-task dataset, using interventions at Layer 15 for consistency with mean-centering. These results are reported in Section 4.2.1, Table 1. We found that FFVs significantly outperformed the other activation steering methods (75.2% vs 18.2% with ActAdd, 45.7% with Mean-Centering) even when injecting it as late as layer 15.
- To directly compare initial FVs with FFVs in solving analogies, we constructed a composite function vector using the initial FVs (IFV-CFV) and measured the effect of replacing them with FFVs. Across the four models tested (GPT-2, GPT-J, LLaMa-2, LLaMa-3.1), adopting FFVs in the CFV rather than the initial FVs improved accuracy by 2-18% for Green’s far-analogy problems and 6.7-10.1% for SAT analogy problems. We present the results in Table 9 of the revised paper.

[1] Todd, Eric, et al. "Function vectors in large language models." (2023).
[2] Turner, Alexander Matt, et al. "Steering language models with activation engineering." (2023).
[3] Jorgensen, Ole, et al. "Improving activation steering in language models with mean-centring." (2023).

---

### Meta-Review · Area_Chair_9g8K · 2026-01-05

**Summary:**

The authors consider *fine-tuning* Function Vectors with a small set of examples, and show this can improve performance considerably on "relation-based word-completion" tasks.

There was a consensus amongst reviewers that while the results do suggest that fine-tuning Function Vectors (on a small set of examples) can improve performance, this is not an especially compelling result (fine-tuning tends to improve performance, after all). In general, the authors do not really adequately motivate the work and analysis here. Moreover, reviewers requested additional experiments (beyond analogy style tasks and comparisons to additional baselines).

**Reviewer Concerns:**

All reviewers took some issue with the framing of the contribution here (namely, what precisely the reader is to takeaway from the fact that fine-tuning improves FV performance for analogy-style tasks). The authors have attempted to clarify this in revision, and this does I think at least partly address the motivation problem. Fundamentally, though, even with the rewrite I'm not sure how compelling it is to show that FVs can be fine-tuned to improve their performance.

The authors added some additional results, in response to requests for evaluation. This is welcome, but still lacks comparison to relevant fine-tuning methods (as highlighted by 177W).

**Reviewer Scores:**

Reading the exchange with P6Px, I'm also not sure that all of the relevant methodological and clarity issues were addressed fully, so may not have changed their score.

Qe3R did not find the reframing compelling, so also seems unlikely to have modified their score.

177W appreciate the reframing so may have updated score marginally.

Unclear if the response would have swayed oDLE.

---

### Decision · Program_Chairs · 2026-01-26

Reject